# Evidence of human infection by a new mammarenavirus endemic to Southeastern Asia

Kim R Blasdell[1,2†], Veasna Duong[1†], Marc Eloit[3], Fabrice Chretien[3], Sowath Ly[1], Vibol Hul[1], Vincent Deubel[1], Serge Morand[4], Philippe Buchy[1,5*]

[1]Institut Pasteur in Cambodia, Phnom Penh, Cambodia; [2]Commonwealth Scientific and Industrial Research Organisation, Australian Animal Health Laboratory, Geelong, Australia; [3]Institut Pasteur, Paris, France; [4]Institut des Sciences de l'Evolution, CNRS, IRD, Université Montpellier, Montpellier, France; [5]GlaxoSmithKline Vaccines R&D, Singapore, Singapore

**Abstract** Southeastern Asia is a recognised hotspot for emerging infectious diseases, many of which have an animal origin. Mammarenavirus infections contribute significantly to the human disease burden in both Africa and the Americas, but little data exists for Asia. To date only two mammarenaviruses, the widely spread lymphocytic choriomeningitis virus and the recently described Wēnzhōu virus have been identified in this region, but the zoonotic impact in Asia remains unknown. Here we report the presence of a novel mammarenavirus and of a genetic variant of the Wēnzhōu virus and provide evidence of mammarenavirus-associated human infection in Asia. The association of these viruses with widely distributed mammals of diverse species, commonly found in human dwellings and in peridomestic habitats, illustrates the potential for widespread zoonotic transmission and adds to the known aetiologies of infectious diseases for this region.

*For correspondence: buchyphilippe@hotmail.com

†These authors contributed equally to this work

## Introduction

Rodents of several species are known hosts of numerous zoonotic pathogens (*Luis et al., 2013*), and are also amongst the peridomestic ensemble that benefit from how humans are modifying the landscape (*Shochat et al., 2006*). Their increased presence will amplify human-rodent encounter rates and opportunities for zoonotic transmission, likely creating an increased risk for human health (*Young et al., 2014*). A concerted effort to establish which human pathogens are present in the environments heavily utilized by humans would therefore be beneficial for the prevention, control and where possible, treatment of such zoonoses.

Mammarenaviruses are predominantly rodent-borne viruses, several of which have been associated with human disease. In Western Africa, Lassa virus (LASV) infection causes an estimated 100,000 to 300,000 infections and approximately 5000 deaths annually (*McCormick et al., 1987*), whilst Junín virus (JUNV) causes regular seasonal outbreaks of viral haemorrhagic fever in Argentina (*Gómez et al., 2011*). So far only Wēnzhōu virus and lymphocytic choriomeningitis virus (LCMV) have been conclusively demonstrated to be present in Asia (*Li et al., 2014*; *Morita et al., 1996*; *1991*). Wēnzhōu virus was identified in China in rodents belonging to four *Rattus* species, *Niviventer niviventer* and in Asian house shrews (*Suncus murinus*), but has not yet been associated with disease. However, LCMV is zoonotic, is primarily hosted by the widespread house mouse (*Mus musculus*), and consequently has the widest known distribution of any mammarenavirus (*Buchmeier et al., 2007*). Human infection with LCMV predominantly results in a relatively mild influenza-like illness

**eLife digest** Rodents have long been notorious for spreading disease among humans. Often the animals can carry viruses and transmit them to humans without becoming ill. Certain species thrive in cities and agricultural areas where they come in close contact with humans; this creates many opportunities to spread infection. As humans urbanize and farm larger swaths of previously wild lands, the risk of rodent-transmitted infections increases. As a result, some scientists are working to identify viruses carried by rodents in human settlements and hopefully prevent them from spreading to humans.

The mammarenavirsuses are a group of rodent-transmitted viruses that commonly cause illness in people in Africa and Latin America. Each year, one such virus – the Lassa virus –sickens as many as 300,000 people in Africa and kills 5,000. So far, only two mammarenaviruses have been found in Asia: one called the Wēnzhōu virus and another called LCMV. However only LCMV is known to cause human illness and many cases of illness caused by mammarenaviruses in Asia may go undetected because they often cause mild symptoms similar to the common cold.

Blasdell et al. have now tested lung samples from 20 species of rodents collected at 7 sites in Cambodia, Thailand, and Laos to look for molecules produced by mammarenaviruses. The tests revealed a strain of Wēnzhōu virus circulating in Cambodian rats that often live in urban areas. A new mammarenavirus was also detected in rodents that live in Thai rice fields. However, infecting wild and domestic rodents with the viruses in the laboratory did not cause many noticeable signs of illness.

Blasdell et al. then tested samples from Cambodian patients who either had influenza-like symptoms or more serious symptoms that are associated with a condition called Dengue fever (which is common in the area). Some patients with respiratory symptoms tested positive for the Wēnzhōu virus. Because the symptoms are mild and similar to those of other common diseases it is likely that the Wēnzhōu virus may be spreading more widely among humans in Asia.

The next challenges are to provide a better estimate of the frequency of this disease in the human population in Asia and to describe the full spectrum of disease that might be associated with this newly discovered infectious disease.

(*Bonthius, 2012*; *Macneil et al., 2012*; *Centers for Disease Control and Prevention (CDC), 2012*). This propensity of some mammarenaviruses to cause mild disease with symptoms common to those found for many other viral infections, illustrates the potential for these infections to go undiagnosed or misdiagnosed. That human infections caused by arenaviruses other than LCMV have not previously been detected in Southeastern Asia (*Charrel and de Lamballerie, 2003*) should therefore not be considered as evidence of absence. As each mammarenavirus is primarily adapted to rodents of distinct species or to closely related rodents of diverse species (*Gonzalez et al., 2007*), the apparent absence of these viruses from a region with high rodent biodiversity (*Pagès et al., 2010*) is surprising. A large proportion of infectious diseases remain undiagnosed in resource poor countries where people rarely visit healthcare professionals and where clinicians often do not have the training or laboratory support for accurate diagnosis (*Mueller et al., 2014*). Here, in the course of a survey for rodent-borne mammarenaviruses, we identified a genetic variant of the recently identified Wēnzhōu virus in Cambodia - provisionally named Cardamones - in rodents belonging to two widespread species, brown rats (*Rattus norvegicus*) and Pacific rats (*R. exulans*), which are commonly found in proximity to humans. We also identified a novel mammarenavirus species in Thailand in rice-field associated greater bandicoot rats (*Bandicota indica*), Savile's bandicoot rats (*B. savilei)* and an Indomalayan niviventer (*Niviventer fulvescens*). In addition, we detected Wēnzhōu virus in several Cambodian patients presenting with fever and respiratory symptoms, indicating that this virus may be causally related with human disease.

## Results

### Identification of two mammarenaviruses in Southeastern Asia

To assess whether mammarenaviruses are present in Southeastern Asian small mammals, we used a previously described RT-PCR (*Vieth et al., 2007*) (*Supplementary file 1A*) to screen 627 homogenised lung samples from small mammals of twenty species (*Supplementary file 1B*) collected from seven sites (two in Cambodia, three in Thailand and two in Laos). Based on the hypothesis of rodent-mammarenavirus interrelation (*Gonzalez et al., 2007*) and the presence of only Old World rodents in the region (*Marshall et al., 1988*), primers targeting Old World mammarenaviruses were used. We detected evidence of mammarenavirus RNA in twenty seven animals from two locations: Veal Renh in Cambodia and Loei in Thailand. Seventeen individuals from two peridomestic species, brown rats (four individuals) and Pacific rats (thirteen individuals), were identified positive in Veal Renh. In Loei, ten individuals from three agricultural-associated species were found positive, namely six Savile's bandicoot rats, three greater bandicoot rats and one Indomalayan niviventer (*Supplementary file 1B*). All but one of the Cambodian positive rodents were collected in settlements, whilst five of the Thai rodents were collected in plantations (*Supplementary file 1C*).

Sanger sequencing of the amplicons indicated the presence of a novel mammarenavirus in the Thai samples. Full genome sequencing was attempted on a single sample from each site: C0649, originally obtained from a wild-caught Pacific rat from Veal Renh, and R5074, originally obtained from a wild-caught Savile's bandicoot rat from Loei. Coding-complete genomes of isolates were acquired by deep sequencing (Illumina HiSeq 2000) of random amplified RNAs from the lungs of Wistar laboratory rats inoculated with either the C0649 or R5074 lung homogenate (C0649: S segment accession KC669696, length 3331 nucleotides (nt); L segment accession KC669690, length 7181nt. R5074: S segment accession KC669698, length 3345nt; L segment accession KC669693, length 7186nt). Primers were designed based on this sequence and in conjunction with a previously published protocol (*Bowen et al., 2000*) (*Supplementary file 1A*), were used to obtain the coding-complete genome of two further isolates: C0617, obtained from a wild-caught brown rat, (S segment accession KC669694, length 3344nt; L segment accession KC669691, length 7171nt) and R4937, obtained from a wild-caught greater bandicoot rat (S segment accession KC669697, length 3379nt; L segment accession KC669692, length 7185nt). Each segment encoded two open reading frames (ORFs) in an ambisense organization with an intergenic region containing a predicted hairpin between the ORFs. Deduced amino acid (aa) and nt sequences from the four isolates were compared to those of other representative mammarenaviruses. An aa sequence divergence of >25% for the nucleoprotein (NP) was found between the Thai isolates and all other known mammarenavirus species, whilst a 3.5–12.7% aa sequence divergence was found between the Cambodian isolates and Wēnzhōu virus (*Table 1*). PAirwise Sequence Comparison (PASC) was performed on both segments for each of the four sequenced viruses. All samples were found most closely related to Wēnzhōu virus, with the Cambodian virus demonstrating 88.5–88.8% identity for the S segment and 86.0–86.3% identity for the L segment, whilst the Thai virus showed 70.3–70.6% identity for the S segment and 62.7–63.1% identity for the L segment (*Table 2*; *Table 2—source data 1*). The International Committee on Taxonomy of Viruses (ICTV) arenavirus species demarcation criteria includes, among others, the association with a main host species or group of sympatric hosts, presence in a defined geographical area, and significant protein amino acid sequence differences, including a difference of at least 12% in the amino acid (aa) sequence of the nucleoprotein to other species in the genus (*King et al., 2012*; *Radoshitzky et al., 2015*). A recent update from the ICTV *Arenaviridae* study group has also recommended the use of the PASC tool for the assessment of novel arenaviruses. Cut-off values chosen for classifying arenaviruses belonging to the same species using this tool are >80% and >76% nucleotide sequence identity in the S and L segments respectively (*Radoshitzky et al., 2015*). The Thai virus is the first mammarenavirus to be detected in *Bandicota* species and, alongside the Cambodian virus is the first to be detected in this geographic region. As this virus also meets nucleoprotein amino acid sequence identities and PASC requirements, we propose that this Thai virus represents a member of a novel species. We suggest to call this novel virus *Loei River mammarenavirus* (after a river close to the site where it was detected) with the abbreviation LORV. Although the Cambodian virus is the first mammarenavirus to be associated with Pacific rats (*R. exulans*) and Cambodia is geographically distant to the Chinese region in which Wēnzhōu virus was originally detected, the sequence homology to Wēnzhōu virus and the association with

**Table 1.** Nucleotide and amino acid sequence identities (%) between Cambodian and Thai isolates and selected other arenaviruses.

| Isolates | Segment or ORF | nt/aa | Cambodian isolates | Thai isolates | Wēnzhōu | Lassa | Ippy | Mopeia | LCMV | Junín | Luna | Morogoro |
|---|---|---|---|---|---|---|---|---|---|---|---|---|
| Cambodian isolates | L segment | nt | 98.9 | 69.2-69.4 | 87.5-88.6 | 55.8-56.4 | 57.7-57.8 | 60.8 | 56.9 | 50.2-50.4 | 61.1-61.4 | 60.9-61.0 |
| | L ORF | nt | 99.3 | 67.3-67.5 | 88.0-89.0 | 59.7-60.6 | 60.4-60.5 | 59.2-59.4 | 55.6 | 50.6-50.9 | 59.6-60.9 | 59.6 |
| | | aa | 99.6 | 69.2-69.4 | 92.2-94.8 | 55.5-56-4 | 57.7-57.8 | 56.6-57.1 | 48.5 | 37.9-38.0 | 55.8-55.9 | 55.6 |
| | Z ORF | nt | 99.5 | 73.4-74.5 | 837.-87.9 | 66.8-67.4 | 63.6 | 62.5 | 57.6-58.2 | 54.9-55.4 | 69.0-70.1 | 64.7-65.2 |
| | | Aa | 98.8 | 79.2 | 89.4-93.9 | 70.1-75.3 | 70.1 | 64.9 | 59.4 | 40.3 | 63.6-64.9 | 61.0 |
| | S segment | nt | 99.5 | 71.7-72.1 | 87.5-89.8 | 61.7-68.1 | 66.6-66.8 | 66.6-67.0 | 61.6-61.8 | 54.6-54.8 | 67.6-68.2 | 66.7-66.9 |
| | NP ORF | nt | 99.3 | 73.1-74.4 | 86.6-90.0 | 67.1-68.2 | 68.3-68.4 | 67.6-67.9 | 62.9-63.2 | 55.1-55.6 | 69.0-70.3 | 68.7 |
| | | aa | 99.8 | 82.9-84.2 | 87.3-96.5 | 72.2-73.8 | 74.4-74.6 | 73.5-73.8 | 64.0 | 51.9-52.5 | 73.3-74.0 | 74.0-74.6 |
| | GPC | nt | 99.7 | 69.1-70.0 | 88.6-89.7 | 67.3-68.6 | 64.7-65.7 | 65.3-65.7 | 61.2-61.4 | 53.4-53.6 | 66.3-66.6 | 49.9-65.2 |
| | | aa | 99.8 | 79.5-81.1 | 95.5-96.4 | 74.2-76.2 | 69.5-71.3 | 71.5-72.8 | 57.2-57.5 | 48.2-43.0 | 72.6-74.4 | 73.1-74.4 |
| Thai virus | L segment | nt | 68-68.1 | 94.6 | 66.6-67.5 | 60.7-61.8 | 61.3-61.5 | 61.1-61.4 | 57.1 | 50.2-50.5 | 60.7-61.4 | 61.4-61.5 |
| | L ORF | nt | 67.3-67.5 | 95.1 | 67.9-68.8 | 59.6-59.9 | 59.9-60.4 | 61.6-62.1 | 55.7-56.2 | 49.9-50.8 | 60.4-61.6 | 59.9-60.2 |
| | | aa | 69.2-69.4 | 96.7 | 69.6-70.7 | 55.7-56.4 | 56.5 | 56.6-57.1 | 49.0-49.4 | 37.1-37.5 | 55.9-56.3 | 56.5-56.7 |
| | Z ORF | nt | 73.4-74.5 | 95.4 | 69.4-75.0 | 65.2-66.8 | 64.1-65.8 | 69.0-69.6 | 58.7-59.2 | 54.3-56.0 | 65.8-72.8 | 66.3-66.8 |
| | | aa | 79.2 | 98.5 | 73.1-74.6 | 67.5-71.4 | 68.8 | 70.1 | 58.4 | 42.9-44.2 | 70.1-71.4 | 66.2 |
| | S segment | nt | 71.7-72.1 | 94.4 | 71.0-72.2 | 65.8-67.3 | 66.1-66.8 | 66.2-67 | 61.7-62.6 | 53.9-55.0 | 65.2-68.1 | 66.1-67.9 |
| | NP ORF | nt | 73.1-74.4 | 94.6 | 72.2-74.1 | 65.2-67.8 | 66.6-67.3 | 66.9-67.7 | 62.4-63.2 | 54.1-54.4 | 66.1-68.0 | 67.3-68.9 |
| | | aa | 82.9-84.2 | 98.1 | 78.3-87.2 | 73.3-74.6 | 72.0-72.9 | 73.8-74.4 | 64.2-64.6 | 49.9-51.0 | 72.0-73.5 | 74-75.5 |
| | GPC | nt | 69.1-70 | 94.1 | 69.2-69.9 | 66.0-67.6 | 65.0-66.1 | 66.4-67 | 61.5-62.6 | 53.9-56.0 | 65.8-69.2 | 66.1-67.5 |
| | | aa | 79.5-81.1 | 97.5 | 80.1-81.1 | 73.5-75.7 | 71.5-72.4 | 71.0-74.2 | 59.7-61.0 | 43.0-43.9 | 74.2-74.8 | 73.1-75.9 |

Note where (L & S gene): Cambodian = KC669690, KC669691& KC669694, KC669696; Thai = KC669692, KC669693 & KC669697, KC669698; Wenhzhou = KM386661, KM051421, KJ909795, KM051420 and KM386660, KM051423, KJ909794, and KM051422; Lassa = GU481076 & GU481077; Ippy = DQ328877 & DQ328878; Mopeia = DQ328874 & DQ328875; LCMV = AY847350 & AY847351; Junín = D10072 & AY216507; ORF = Open reading frame; NP = Nucleoprotein; GPC = Glycoprotein; nt = Nucleotide; aa = Amino acid; LCMV =: Lymphocytic Choriomeningitis Virus

rodents of a common host species (*R. norvegicus*), indicates that the Cambodian virus represents a genetic variant of *Wēnzhōu mammarenavirus* (*Li et al., 2014*). We propose to provisionally name this variant Cardamones (after the chain of mountains near Veal Renh).

In maximum likelihood phylogenetic analyses, the Southeastern Asian viruses, along with the Chinese isolates of Wēnzhōu virus, formed an independent clade within the Old World mammarenaviruses. The Cambodian virus sequences clustered with the Wēnzhōu virus sequences, whilst LORV sequences clustered independently. This Asian mammarenavirus clade formed a sister clade to Ippy

**Table 2.** Summary of PASC analysis.

| Sample ID | Country of origin | PASC: S segment | | PASC: L segment | |
|---|---|---|---|---|---|
| | | Sequence identity (%) | Closest virus | Sequence identity (%) | Closest virus |
| C0617 | Cambodia | 88.80 | Wēnzhōu virus | 86.27 | Wēnzhōu virus |
| C0649 | Cambodia | 88.51 | Wēnzhōu virus | 85.98 | Wēnzhōu virus |
| R4937 | Thailand | 70.63 | Wēnzhōu virus | 62.71 | Wēnzhōu virus |
| R5074 | Thailand | 70.29 | Wēnzhōu virus | 63.08 | Wēnzhōu virus |

Source data 1. PASC analysis.

virus (IPPYV) in both full RNA-dependent RNA polymerase (L) and NP gene analysis, and to the African Lassa-related viruses as a whole in glycoprotein precursor (GPC) gene analysis (*Figure 1a* and *Figure 2*).

## Experimental infections in rodents

To establish if infection and transmission of the Cardamones variant of Wēnzhōu virus could be replicated in rodents of two of the suspect host species, experimental infections were performed. Despite several attempts to obtain a culture isolates in tissue culture, including multiple passages, we were unable to propagate the Cambodian isolates in the cell lines available in our laboratory (Vero E6, MDCK, BHK), or in primary cell lines from lungs obtained from a Wistar laboratory rat. Experimental infections were therefore conducted using homogenised lung tissue from wild-caught RNA-positive animals to infect both Wistar laboratory rats and wild-caught or first generation Pacific rats captured in the Mondulkiri site. Male and female adult animals were used. Due to logistical limitations, parallel experiments were not conducted for LORV. Of 85 animals inoculated intraperitoneally with the Cardamones variant of Wēnzhōu virus-positive homogenate, 77 (64/68 Wistar laboratory rats; 13/17 Pacific rats) were successfully infected as measured by detection of viral RNA by RT-PCR in organs and/or seroconversion by IFA and/or ELISA (*Supplementary file 1D*). Organs positive for Wēnzhōu virus (i.e. brain, heart, lung, liver, spleen, kidney, bladder, peritoneum) varied between individuals (data not shown), with individuals tested usually demonstrating a generalised infection. Organs from randomly selected adult individuals collected between 3dpi and 22dpi or 28dpi for Wistar laboratory rats and Pacific rats respectively, were subjected to quantitative real-time RT-PCR (qRT-PCR) and copy number was determined. Copy numbers increased over time with high copy numbers ($10^8$–$10^{11}$ cDNA copy/mg) detected in several organs from 7dpi in Pacific rats and 11dpi in Wistar laboratory rats. Specifically, high copy numbers were detected in the liver, spleen and lung of most individuals, with the latter suggesting the potential for respiratory transmission. Mammarenavirus RNA was detected in most inoculated Wistar laboratory rats killed between 3-56dpi (52/68 individuals) and Pacific rats killed between 3-37dpi (11/17 individuals), although no animals were tested beyond these time points. Mammarenaviruses are known to cause both acute and chronic infections in their rodent hosts [17] and the duration of infection observed here is comparative to that seen in experimental infections of the natural hosts of other mammarenaviruses (*Fulhorst et al., 1999*; *Walker et al., 1975*). However, as lung homogenates were used instead of titrated virus, the variability of the dose received by each animal may have impacted on the infection course, likely resulting in the variations and inconsistencies observed (*Childs et al., 1993*).

In both species, we observed horizontal transmission, but only between pups and their dams (*Supplementary file 1D*). Vertical transmission could not be confirmed for either species. Mammarenavirus RNA was detected in a single pup from each of the two Pacific rat litters but all other pups from these litters were negative, and antibodies detected in additional pups may have been of maternal origin. LCMV infection of house mice must occur shortly before or after mating in order for dams to transmit the virus to a high proportion of their pups (*Skinner and Knight, 1974*). Therefore the absence of vertical transmission observed here could have been due to the late stage of gestation at which most dams were inoculated (*Supplementary file 1D*), although vertical transmission has not been confirmed in all mammarenaviruses (*Mills et al., 2007*).

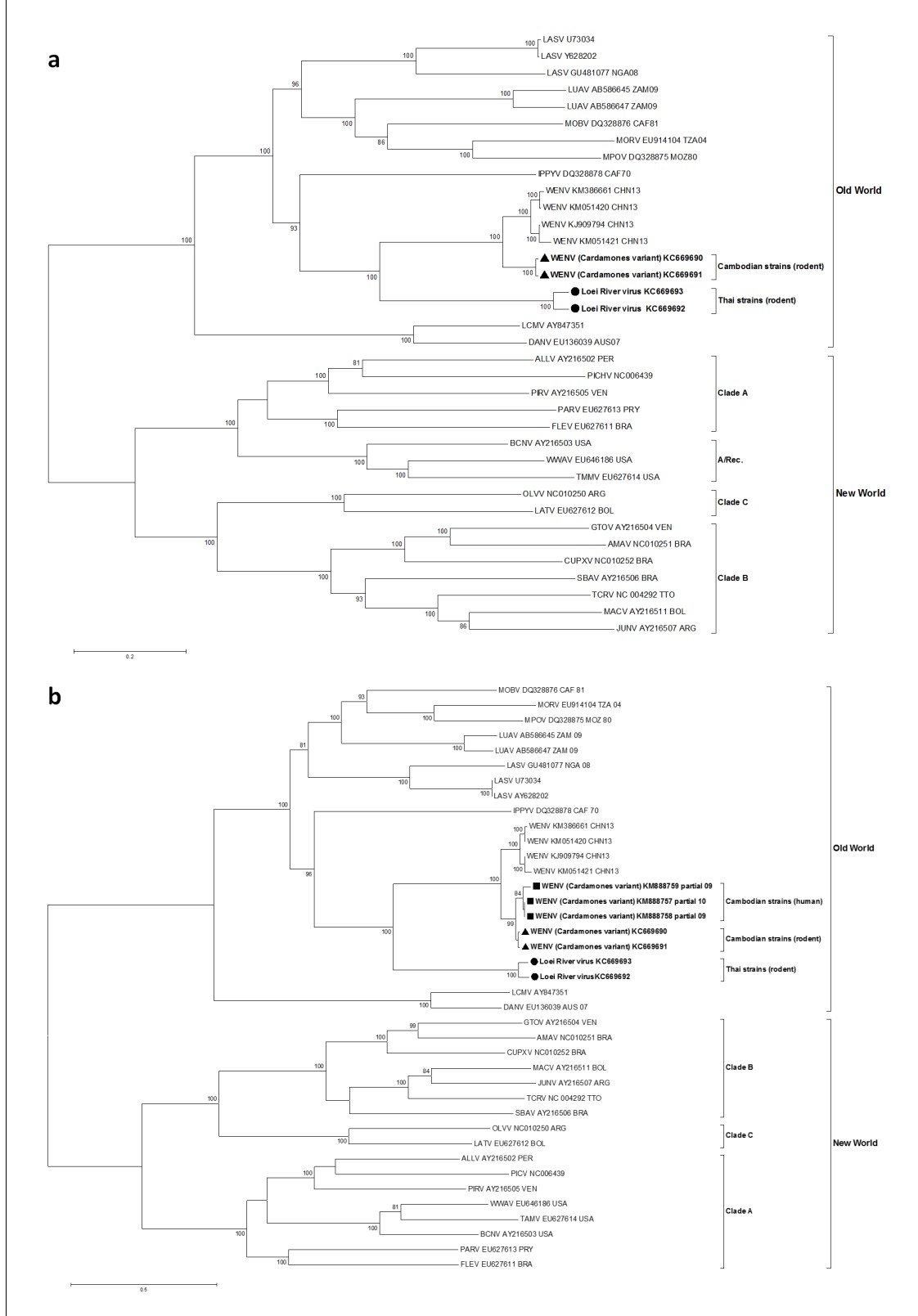

**Figure 1.** Maximum likelihood phylogenetic tree of novel arenavirus isolates and other representative arenaviruses for a, the complete ORF of the L gene with sequences from rodents only, b, partial L sequences including sequences from rodents and patients. Cambodian strains detected in rodent (triangle), human (square) and Thai strains detected in rodent (circle) are in bold. Clade A, B and C are three evolutionary lineages of New World

*Figure 1 continued on next page*

*Figure 1 continued*

arenaviruses within the Tacaribe complex. A/Rec denotes the recombinant clade including the three Northern American viruses. The virus names are in abbreviation according to *Radoshitzky et al. (2015)*.

Clinical signs are rarely seen in mammarenavirus rodent reservoirs, although more subtle signs including reduced weight and body mass index have been observed (*Borremans et al., 2011*). To establish if Wēnzhōu virus infection of rodent hosts results in lesions, histopathological analysis and chromogenic immunohistochemistry were performed on tissues from two Pacific rats inoculated intra-peritoneally with lung tissue homogenates obtained from a wild rat that tested positive for Cardamones variant of Wēnzhōu virus. Electron microscopy clearly identified virus particles with a specific mammarenavirus-type morphology (*Figures 3a and b*) in the lung tissues. Histopathological examination of the lungs revealed only severe diffuse pneumonia (*Figures 3c–h*). Other organs were not affected except for vascular congestion, probably due to pre mortem conditions. This relatively limited pathology is in congruence with the suspected reservoir status of this species.

## Human mammarenavirus infections in Cambodia

The peridomestic nature of the mammalian hosts of Wēnzhōu virus (*Ivanova et al., 2012*), indicates that this virus would have ample opportunities to transmit to humans. We therefore decided to establish if zoonotic transmission does occur and if it is associated with clinical disease by testing human clinical samples from eight different sources (*Table 3*). In group 1, a total of 89/510 patients presenting with dengue-like and influenza-like syndromes but who tested negative for dengue and influenza infection (17.4%) tested positive for anti-arenavirus IgG. The mean age of positive patients was 13 years (95% confidence interval (CI): 10.4–15.6; range: 4 months to 70 years) which was not statistically different (p=0.06) from negative patients (mean age: 10.6 years; 95% CI: 9.6–11.6; range: 7 months to 65 years). The highest antibody prevalence was detected in the 6 to 10 year old age group (40.5%). There was no difference in the sex ratio between seropositive and seronegative patients (49.5% and 43%, respectively; p=0.29). The seroprevalence was particularly high in some villages of Kampong Cham province (*Figure 4*). Seroconversions were observed in 7 young patients with undocumented fever or influenza-like illness (*Table 4*). These patients had a mean age of 8.1 year (95% CI: 4.3–12; range: 3–16 years) with a sex ratio of 0.71, which were not significantly different to mammarenavirus negative patients with paired sera (mean age: 9.6 years; 95% CI: 8.2–10.9; range: 7 months to 65 years; p=0.67; sex ratio: 0.45; p=0.17). Intervals between the collection of the acute and the convalescent serum samples for these patients ranged between 2 and 15 days with a mean of 7.4 days (*Supplementary file 1E*). Retrospective clinical data were available for two of these patients. The first case (S1) was a 3 year old boy who presented with fever (38°C), rhinorrhea and nausea. The second case (S2) was a 9 year old boy presenting with mild fever (37.5°C), headache, cough, rhinorrhoea, and nausea. Muscle pain, joint pain, rash and bleeding were absent in both cases. Only fever (37.5°C to 38.9°C) was recorded in the five remaining cases (S3-7). Of note, self-treatment with antipyretics is extremely common in Cambodia and therefore the temperature measured at the time of medical examination is often normal. In addition to clinical patients, 529 samples from healthy individuals from Kampong Cham province who participated in a community-based dengue seroprevalence study (group 2) were also screened for anti-mammarenavirus IgG antibodies and 70 (13.23%) were found positive (*Table 3*). The mean age of positive individuals was 10.71 (95% CI: 9.76–11.67; range: 1.1–19 years) with a sex ratio 0.47. The positivity rate and mean age of positive cases in healthy individuals compared to patients presenting with dengue-like and influenza-like symptoms were not significantly different (p=0.059 and p=0.09, respectively). A limited number of human samples (Total = 372, *Supplementary file 1F*) were tested by both mammarenavirus IFA and IgG ELISA, demonstrating in general consistent results (coefficient of correlation of 0.565 with a p<0.001). The comparison those two assays is only indicative as the tests were used for different purposes, due to different properties. IFA was chosen for screening purposes because of the ability of the test to detect antibodies directed against a wide range of arenaviruses while the ELISA was developed with the nucleoprotein of the Cardamones variant of Wēnzhōu virus is order to increase the specificity of the serology. Indeed, the Wēnzhōu virus IgG ELISA was tested against LCMV positive human sera, the only other zoonotic mammarenavirus currently presumed to be present in

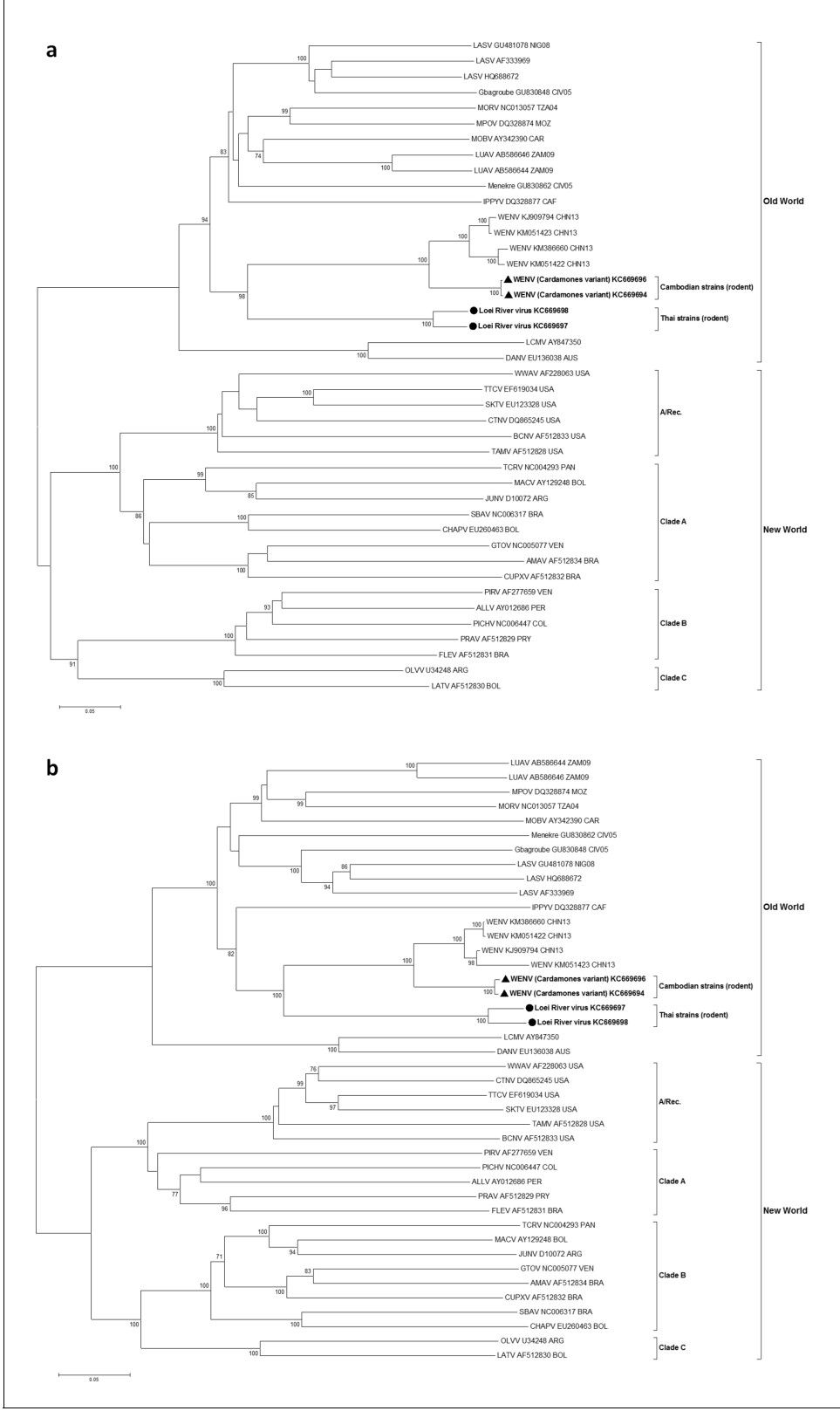

**Figure 2.** Maximum likelihood phylogenetic tree of novel arenavirus isolates and other representative arenaviruses for a, the complete ORF of GPC gene and b, complete ORF of NP gene. Cambodian strains (triangle) and Thai strains (circle) detected in rodents are in bold. Clade A, B and C are three evolutionary lineages of New World arenaviruses within the Tacaribe complex. A/Rec denotes the recombinant clade including Northern American viruses. The virus names are in abbreviation according to *Radoshitzky et al. (2015)*.

Southeastern Asia, but no serological cross-reaction was observed. The IgG ELISA seemed to be slightly more sensitive than the mammarenavirus IFA, detecting 11 additional positive samples (*Supplementary file 1F*). The mammarenavirus IFA identified 2 samples found negative by the IgG ELISA as positive. However these infections may have been caused by more distantly related viruses, such as LCMV, not capable of detection by the more specific IgG ELISA. Despite this, as mammarenaviruses are notoriously serologically cross-reactive by ELISA (*Fukushi et al., 2012*) it is unlikely that this assay is perfectly specific to Wēnzhōu virus. Instead it is likely that this test would be capable of detecting other closely related arenaviruses, including novel species. However antisera to other arenaviruses were not available for testing. As for most serological assays, the precise causative agent in these seropositive patients can therefore not be identified conclusively.

The remaining six groups were tested by a 'screening' semi-nested RT-PCR derived from the RT-PCR method previously described by Vieth et al. (*Supplementary file 1A*) or by the qRT-PCR developed in this study. All the 200 patients who presented with signs and symptoms of meningo-encephalitis (group 3) and tested negative by PCR and serology for the main etiological agents of central nervous system infections, tested negative for mammarenavirus in the cerebrospinal fluid. No trace of mammarenavirus RNA was detected in any of the 253 sera collected during the acute febrile phase of patients presenting with possible dengue fever or dengue hemorrhagic fever (group 4), and who tested negative for dengue virus infection. But mammarenavirus RNA was detected in the respiratory specimens of 6 of 999 (0.6%) Cambodian patients from groups 5 and 6, who presented with either mild (ILI: influenza-like illness) or more severe (ALRI: acute lower respiratory infection) respiratory symptoms respectively (*Table 3* and *Table 4*).

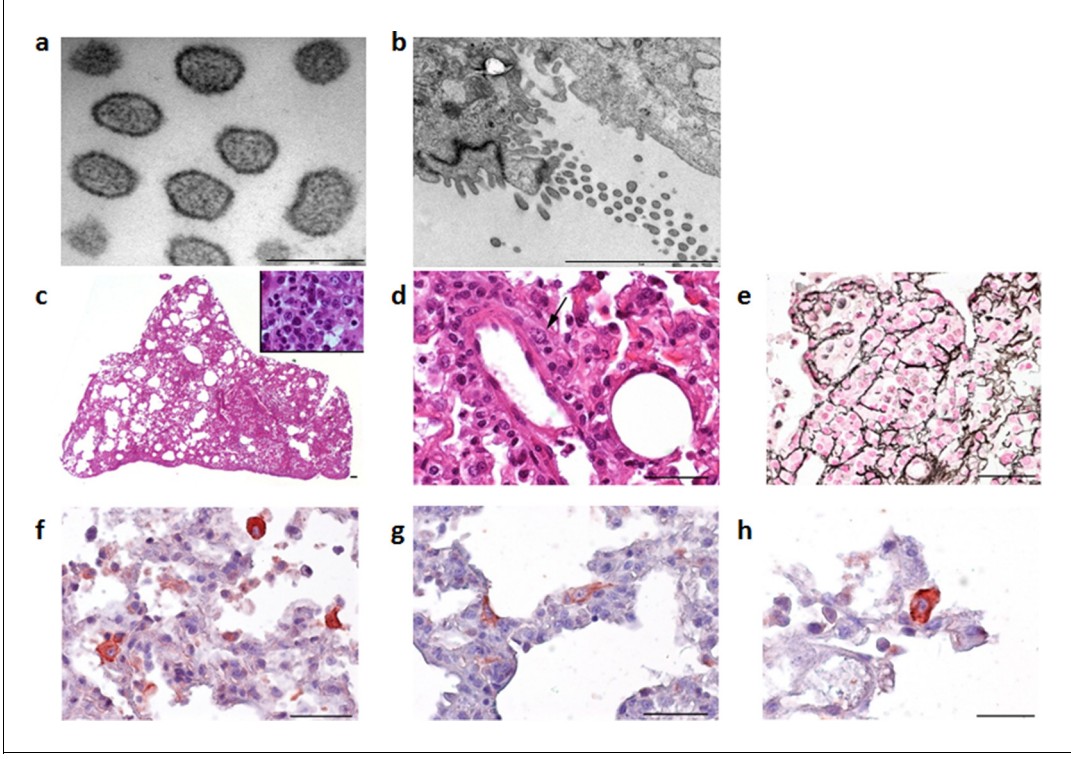

**Figure 3.** Electron microscope images of the Cardamones variant of Wēnzhōu virus in the rat lung tissues, a (scale: 200 nm) & b (scale: 2 μm). Histopathological examination of the lungs revealed only severe diffuse pneumonia, with lesions associated with acute exudative inflammation characterised by foci of consolidation surrounded by extremely rare aerated parenchyma, c, vascular congestion, acute bronchiolitis and diffuse leukocytic infiltration with lymphocytes, polymorphonuclear and macrophages (suppurative exudate in the lumen and parietal inflammation), c and d. Reticulin staining demonstrated the severe destruction of lung parenchyma, e. Chromogenic immunohistochemistry identified inflammatory foci with numerous positive cells, f, primarily inflammatory cells including macrophages, but in the more preserved parenchyma and aerated parenchyma some epithelial (pneumocytes), g, and alveolar macrophages, h, were also clearly stained.

**Table 3.** Details of patient samples tested.

| Group | Clinical signs | Sample type (number of samples) | Collection dates | Collection locations | Mean age in years (range) | 95% CI (years) | Sex ratio | Test used | Number (proportion) of positive |
|---|---|---|---|---|---|---|---|---|---|
| 1 | Dengue-like/ influenza-like illness | Acute and convalescent sera (n=510 including 98 acute, 214 convalescent and 198 paired sera) | 2005-2010 | Kampong Cham, various | 11.0 (4 months to 70 years) | 10.0-12.0 | 0.44 | IgG ELISA | 89 (17.4%) - Acute sera: 33 (33.7%) - Convalescence sera: 25 (11.7%) - Paired sera: 31 (15.7%) with evidence of seroconversion in 7 pairs (3.5%) |
| 2 | Healthy individuals (community dengue seroprevalence study) | Sera (n=529) | 2009 | Kampong Cham | 10.0 (1 month-20 years) | 9.59-10.42 | 0.50 | IgG ELISA | 70 (13.23%) |
| 3 | Meningo-encephalitis | Cerebrospinal fluid (n=200[£]) | 2013-2014 | Various | 6.5 (3 months-15 years) | 5.95-7.06 | 0.40 | Real Time RT-PCR | 0 (0%) |
| 4 | Dengue-like febrile illness | Sera collected during febrile stage (n=253*) | 2009, 2011-2013 | Various | 8.2 (1 to 38 years) | 7.6-8.8 | 0.48 | Semi-nested RT-PCR | 0 (0%) |
| 5a | Influenza-like illness (negative for four common respiratory viruses) | Nasopharyngeal swabs (n=328) | 2007-2012 | Various | 13.2 (1 month to 83 years) | 12.0-14.4 | 0.51 | Semi-nested RT-PCR | 4 (1.2%) |
| 5b | Influenza-like illness (positive for four common respiratory viruses) | Nasopharyngeal swabs (n=392[#]) | | | | | | | 0 (0%) |
| 6 | Acute lower respiratory infection | Nasopharyngeal swabs (n=279[$]) | 2008-2009 | Various | 2.7 (7 months to 63 years) | 2.1-3.4 | 0.41 | Semi-nested RT-PCR | 2 (0.7%) |
| 7 | Healthy individuals (H5N1 contacts) – negative control | Nasopharyngeal swabs (n=266[§]) | 2005-2011 | Various | 29.3 (1 to 79 years) | 27.1-31.5 | 0.45 | Semi-nested RT-PCR | 0 (0%) |
| 8 | Healthy anti-rabies vaccination volunteers – negative control | Nasopharyngeal swabs (n=238[¥]) | 2013 | Institute Pasteur Cambodia (Phnom Penh) | 26.7 (9 months to 83 years) | 24.5-28.8 | 0.56 | Semi-nested RT-PCR | 0 (0%) |

CI = Confidence Interval

[£] 3 no information on age; * 8 no information on age and sex; [#] 2 no information on age and sex; [$] 1 no information on age; [§] 5 no information on age and sex; 51 no information on sex; [¥] 5 no information on age and sex

In group 5a, the first case was a 3 year-old female with no known underlying disease, sampled 1 day after the onset of fever associated with cough and rhinorrhea. A co-infection with human parainfluenza virus 1 (HPIV-1) was also detected. Case 2 was a 9 year-old male with no known underlying disease sampled 2 days after onset of fever associated with cough, rhinorrhea, nausea, vomiting and severe headache. Case 3 was a 31 year-old male farmer, who presented 2 days after the onset of fever associated with productive cough, rhinorrhea, nausea and severe headache. Case 4 was a 45 year-old female farmer sampled 1 day after onset of fever associated with productive cough, rhinorrhea, nausea, severe headache and muscle pain. None of these 4 patients reported other signs and symptoms often associated with influenza (otalgia, sore throat) or dengue (rash, retro-orbital pain, joint pain, bleeding). Hospitalization was not required in any of these cases. In group 6 the first case

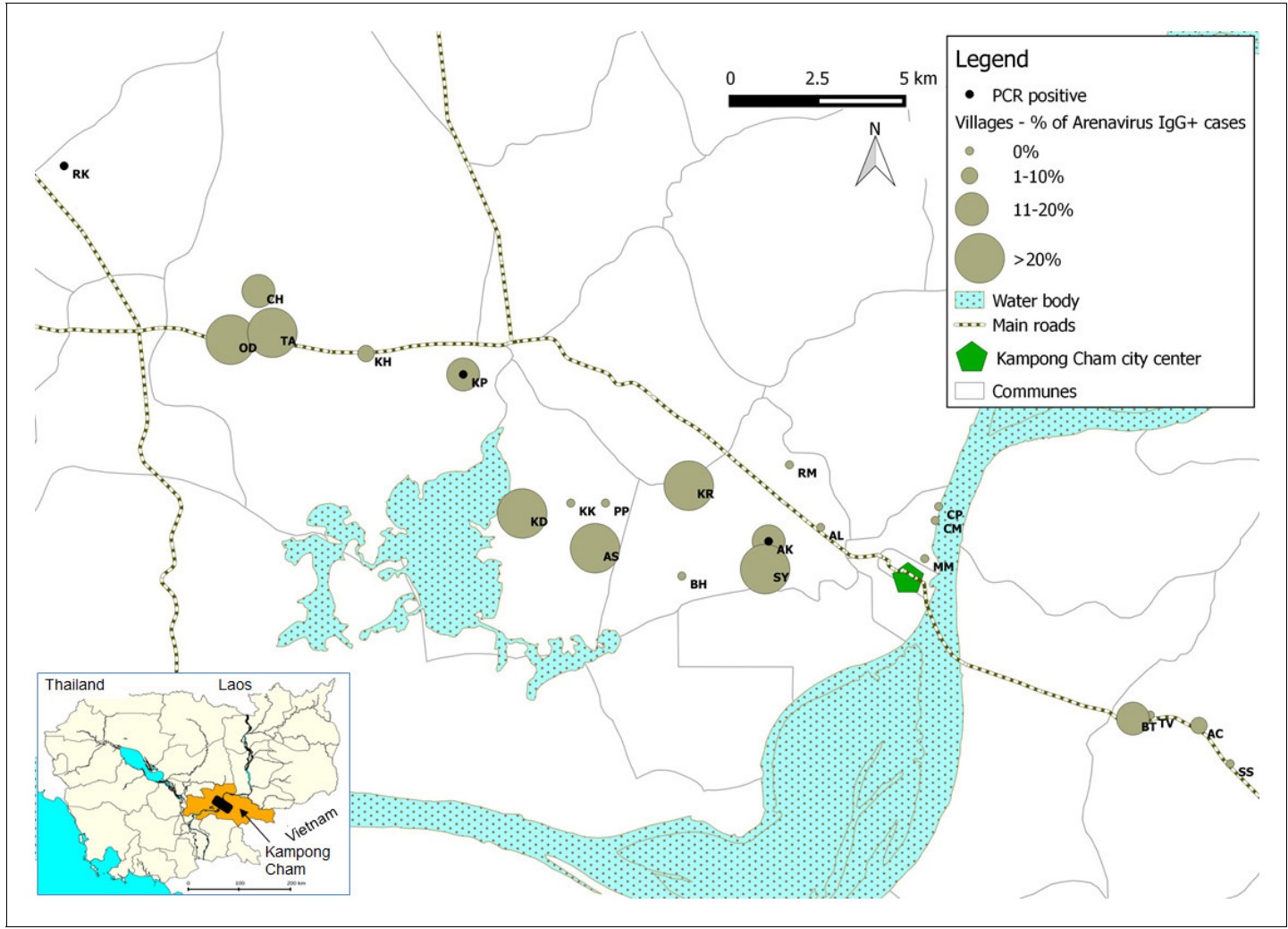

**Figure 4.** Representation map of the percentage (%) of patients who tested positive by anti-arenavirus IgG ELISA and of 6 patients who tested positive by L gene RT-PCR in the villages from Kampong Cham province. Village names: AC=Andoung Chea, AK=Ampil Kraom, AL=Ampil Leu, AS=Andoung Svay, BH=Banteay Thma, BT=Boeng Tras, CH=Chachak, CM=Chong Thnal Muoy, CP=Chong Thnal Pir, KD=Kdei Boeng, KH=Kakaoh, KK=Kouk Kream, KP=Krasang Pul, KR=Krala, MM=Memay, OD=Ou Da, PP=Prey Phdau, RK=Roung Kou, RM=Romeas, SS=Srae Siem, SY=Sya, TA=Tuol Ampil and TV=Tuol Vihear.

(case 5) was that of an 8 month-old female with a recent history of fever (<4 days) who presented to hospital with fever (38°C), productive cough, cardiac frequency at 130bpm, respiratory rate at 46/min, dyspnea and wheezing. The blood cell count performed at admission demonstrated anaemia (haemoglobin: 93 g/L), a normal white cell count and formula (leucocytes: $12.6 \times 10^9$/L, lymphocytes: $5.2 \times 10^9$/L, neutrophils: $5.7\ 10^9$/L). There was no liver cytolysis (SGOT: 46 U/L; normal value: 5–40 U/L, SGPT: 33 U/L; normal value: 5–50 U/L), the creatinine was normal (40 µmol/L) as was glycaemia (5.0 mmol/L). The chest radiograph which was retrospectively reviewed by an expert pulmonologist was normal and the final clinical diagnosis was of an acute bronchiolitis of probable viral origin. The patient received amoxicillin, gentamicin and bronchodilators for 4 days and was discharged after full recovery. The second hospitalized case (case 6) was a 3 month-old boy, admitted with a history of fever but with a normal body temperature at the time of medical examination, associated with productive cough, cardiac frequency of 132bpm, respiratory rate of 50/min, dyspnea and wheezing. The blood cell count (haemoglobin: 112 g/L, leucocytes: $11.6 \times 10^9$/L, lymphocytes: $7.2 \times 10^9$/L, neutrophils: $2.4 \times 10^9$/L) and renal function (creatinine: 47 µmol/L) were normal for the patient's age. A hypoglycaemia of 2.8 mmol/L was reported. SGOT and SGPT were at 53 U/L and 44 U/L, respectively. The

**Table 4.** Clinical details of human arenavirus infections.

| Case no. | Test used | Sex | Age | Hospitalised | Fever | Cough | Rhinorrhea | Nausea/ vomiting | Severe headache | Muscle pain | Other symptoms | Co-infection |
|---|---|---|---|---|---|---|---|---|---|---|---|---|
| 1 | Screening nested RT-PCR | Female | 3 years | No | Yes | Yes | Yes | | | | | Human para-influenza virus 1 |
| 2 | Screening nested RT-PCR | Male | 9 years | No | Yes | Yes | Yes | Yes | Yes | | | |
| 3 | Screening nested RT-PCR | Male | 31 years | No | Yes | Yes | Yes | Yes | Yes | | | |
| 4 | Screening nested RT-PCR | Female | 45 years | No | Yes | Yes | Yes | Yes | Yes | Yes | | |
| 5 | Screening nested RT-PCR | Female | 8 months | Yes | Yes | Yes | | | | | Dyspnea, wheezing, moderate anaemia (93g/L) | Unspecified Rhinovirus |
| 6 | Screening nested RT-PCR | Male | 3 months | Yes | Yes | Yes | | | | | Dyspnea, wheezing, hypoglycaemia (28mmol/L) | Unspecified Rhinovirus |
| S1 | ELISA | Male | 3 years | No | Yes | | Yes | Yes | | | | |
| S2 | ELISA | Male | 9 years | No | Yes | Yes | Yes | Yes | Yes | | | |
| S3-7 | ELISA | | | No | Yes | | | | | | | |

independent expert pulmonologist who reviewed the chest radiograph retrospectively described patchy infiltrates and confirmed the initial diagnosis of acute bronchiolitis of probable viral origin made by the hospital paediatrician. The patient fully recovered and was discharged after 3 days of treatment with bronchodilators, amoxicillin and gentamicin. In both of the hospitalised cases a rhinovirus was also detected in the nasopharyngeal swab samples, but it should be noted that rhinovirus infection is not always associated with symptomatic disease (*Buecher et al., 2010*).

The association between the respiratory illness and the detection of the mammarenavirus was statistically significant ($p<0.05$) (*Supplementary file 1G*). Nevertheless, probably because of the limited sample sizes, the difference remained statistically significant only in adults over 30 years of age when the analysis was stratified by age groups (*Supplementary file 1H*). Direct sequencing of the 302bp amplicons of L gene obtained from three of these six respiratory specimens identified Wēnzhōu virus (*Figure 1b*) with a 98–99% similarity to the strain detected in rodents from Veal Renh and 88.9–91% with Wēnzhōu virus isolated in China. Based on the absence of contamination of negative controls, and because 2 to 10 nucleotides differences were identified between the patient sequences and those of the positive control of rodent origin (*Supplementary file 1I*), the positive patient samples are extremely unlikely to be a result of cross-contamination. In addition, the respiratory samples from 2 patients who tested positive by the screening nested RT-PCR were also tested by a qRT-PCR that targets a totally different region of the L gene (*Figure 5* and *Supplementary file 1A*) and both samples tested positive with a viral load of 200 and 350 equivalent cDNA copies per reaction, respectively (or 17,900 and 30,600 copies/ml of sample in virus transport medium). All the 6 patients were additionally tested for 17 common respiratory viruses by RT-PCR (*Buecher et al., 2010*) and co-infections were detected in three out of six patients presenting with respiratory symptoms. The clinical data recorded for these patients (cases 1–6) are described in *Table 4*. All the human mammarenavirus cases detected by molecular methods originated from Kampong Cham province. Although a total of 19 villages of the Kampong Cham province were included in the community-based study that comprised group 5, all the 4 cases detected in sub-group 5a originated from Krasang Pul village (KP) and were obtained between January and July 2009. Of note, patients from KP were over-represented (32%) amongst the total number of patients tested in sub-group 5a. The two individuals found mammarenavirus positive in group 6 were from two other villages, Roung Kou (RK) and Ampil Kraom (AK), in Kampong Cham province, both of which are situated approximately 8 km distant from KP (*Figure 4*). This, in conjunction with the relatively high mammarenavirus seropositivity rates also observed in this province (*Figure 4*), suggests that another focus of Wēnzhōu virus exists here.

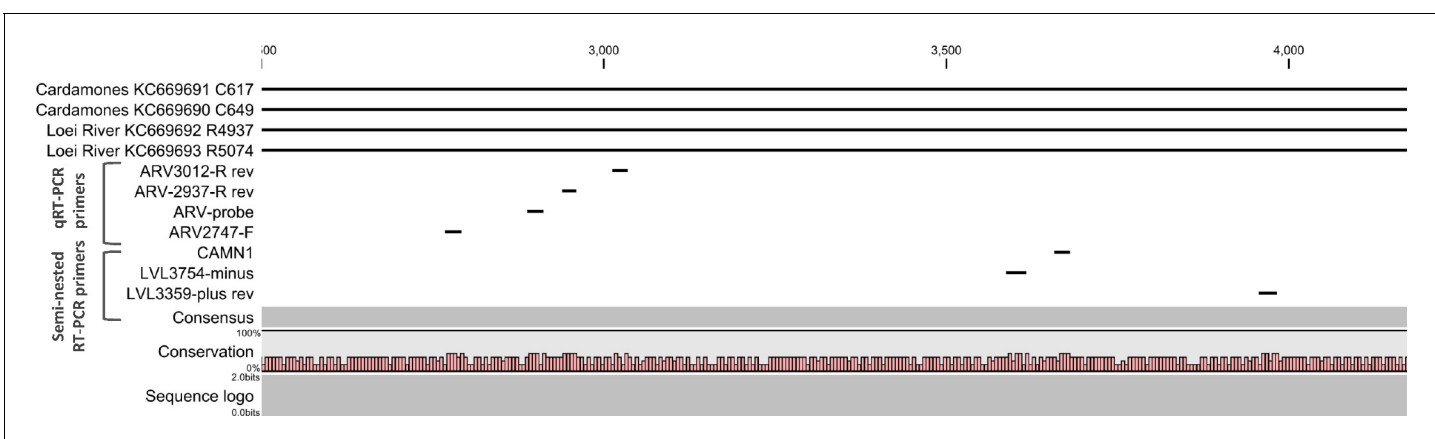

**Figure 5.** Positions of primers used in diagnostic PCR assays. The positions of the primers used for qRT-PCR are included in their names and these positions are based on the sequence of the L gene of Cardamones variant of Wēnzhōu virus. The positions of the primers used in the nested RT-PCR originate from the article of Vieth et al. (*Vieth et al., 2007*).

## Discussion

The identification of a novel mammarenavirus in Thai rodents and of a genetic variant of Wēnzhōu virus in Cambodian rodents is the first unequivocal evidence of mammarenaviruses in Southeastern Asia. The identification of the latter virus in Cambodia in individuals of *R. exulans* sympatric to infected individuals of the known host, *R. norvegicus*, expands both the geographic and host range of this mammarenavirus species. Although LCMV was not identified in this study (probably due to the absence of its host the house mouse from large regions of Southeastern Asia (*Aplin et al., 2003*) and the study sites in particular) it is also proposed to have an Asian origin (*Albariño et al., 2010*). Despite this shared geographic heritage, both Wēnzhōu virus and LORV appear to be more closely related to the African LASV/IPPYV group viruses. This is also surprising from the viewpoint of the rodent-mammarenavirus co-evolution hypothesis. Based on this hypothesis a closer relationship would be expected between LASV/IPPYV and LCMV, since the rodent hosts (*Arvicanthis, Mastomys* and *Praomys* species) (*Childs et al., 1993*; *Lecompte et al., 2005*) of the African mammarenaviruses are more closely related to *Mus* spp. than they are to the Asian Rattini tribe (*Lecompte et al., 2008*). As such this finding alongside the recent discovery of divergent arenaviruses in boid snakes (recently classified as reptarenaviruses, the sister genus to mammarenaviruses) and more in depth analyses of rodent-mammarenavirus relationships, suggest that the history of the family *Arenaviridae* is more complex and the theory of co-evolution less robust than initially thought (*Coulibaly-N'Golo et al., 2011*; *Irwin et al., 2012*; *Stenglein et al., 2012*).

Although the host status of rodents of several *Rattus* spp. could be called into question based on the pathologies observed in some animals experimentally infected with Cardamones variant of Wēnzhōu virus, detrimental effects of infection have been observed in the natural host of another mammarenavirus, Mopeia virus (*Borremans et al., 2011*), and in the current study may have resulted from the unnatural route of infection. In addition, the successful experimental infection and transmission of Wēnzhōu virus in rodents of two of the proposed host species, alongside the detection of this virus in rodents of several *Rattus* spp. from two geographically remote locations (*Li et al., 2014*), support their status as the reservoir hosts. Due to the relative host specificity of mammarenaviruses, confirmation of the host species of each of these viruses has important implications for their geographic distributions. Many of the species in which Wēnzhōu virus or LORV have been detected are widely distributed (*Aplin et al., 2003*; *Ruedi et al., 1996*; *Wodzicki and Taylor, 1984*). Rodents of these species are also associated with either agriculture (*Bandicota* spp.) or urban settings (*Rattus* spp.) (*Aplin et al., 2003*) and as such are likely to be among the few species that may benefit from urbanisation and agricultural intensification. This could ultimately result in increases in both the distribution of their viruses and in their opportunities for zoonotic transmission.

Our results suggest that the ecology of both viruses, through their association with peridomestic rodents of several species, mirrors that observed for known zoonotic mammarenaviruses. Like LORV, JUNV is associated with agricultural pest species (*Busch et al., 2000*), whilst Wēnzhōu virus, LASV and LCMV are all hosted by rodents with a preference for human dwellings (*McCormick et al., 1987*; *Pocock et al., 2004*). This association with highly utilised human habitats indicates that humans are potentially exposed to both viruses on a regular basis as suggested by the relatively high seroprevalence observed in both the patient and the control group. Therefore, the identification of human Wēnzhōu virus infection is perhaps unsurprising from an epidemiological viewpoint, but nevertheless, represents the first detection of human mammarenavirus infection in Asia. The results from this study suggest that human infection with Wēnzhōu virus, or a closely related virus, occurs widely throughout Cambodia, and may possibly result in disease in some cases.

Confirmation of causality however is made problematic by the mild and indistinctive clinical presentation observed, the detection of co-infecting pathogens capable of producing a similar clinical picture in three out of six cases (although rhinovirus infection is not always symptomatic) (*Buecher et al., 2010*) and the limited number of cases. Although virus was never detected in healthy individuals and all patients with confirmed Wēnzhōu virus infections experienced similar symptoms of acute febrile bronchitis, the non-detection of co-infecting agents in three cases despite extensive testing does not mean that other untested or unknown pathogens were not present. However the clinical picture observed was similar to that reported for milder LCMV infection (*Baum et al., 1966*; *Knust et al., 2014*). If like LCMV, Wēnzhōu virus rarely causes distinctive severe

disease, then despite the high apparent incidence of the infection indicated by the retrospective serological study, causality is unlikely to be confirmed without further surveys.

Although our findings indicate that novel mammarenaviruses are present in Southeastern Asia and that humans appear to be regularly infected in the region, there are several issues that still need to be addressed. A major one was our lack of success in isolating both Cardamones variant of Wēnzhōu virus and LORV in tissue culture. This was surprising as samples clearly contained viable virus as evidenced by successful experimental infection of rats. It was also inhibitive to downstream analyses, as a successful method of isolation could greatly aid in the development of serological tests and in confirming proof of causality in suspected cases of human disease. The reasons behind our inability to isolate these viruses are unknown. It is possible that the viruses may have replicated too slowly to be detected by the methods used but this seems unlikely based on the successful isolation of other arenaviruses under similar conditions (*Gómez et al., 2011*). More likely is the possibility that samples contained viral titres that were too low for successful isolation in the cell lines tested (*Lednicky et al., 2012*) or that cell lines were contaminated with Mycoplasma. Isolation attempts were made in cell lines previously used to culture other mammarenaviruses (e.g. Vero, MDCK, BHK), but as Wēnzhōu virus was isolated in canine macrophage DH82 cell line (*Li et al., 2014*), it would be prudent to attempt isolation in this cell line in the future. A serological test highly specific to Wēnzhōu virus is also needed. As such, the specificity of the ELISA test developed during this study needs to be assessed against other mammarenaviruses and potentially optimised further, or an alternative strategy employed. In order to strengthen the case for Wēnzhōu virus as an agent of human disease, future surveys would ideally include a comparison of the seroprevalence in exposed and non-exposed populations, using such a Wēnzhōu virus-specific test. Although attempts were made during this study to obtain sera from individuals not native to the Southeastern Asian region, only limited samples were available. Many of these were from individuals who had been resident in this region for some time and therefore may have been exposed to Southeastern Asian mammarenaviruses. Sourcing and testing of sera from non-endemic areas is therefore needed. Studies aimed at detecting seroconversion in conjunction with molecular testing for viral RNA are also required, with the ultimate goal of isolating the virus in tissue culture or rodents. If Wēnzhōu virus does cause disease, it is also possible that like LASV and LCMV, more severe disease may occur in particular individuals such as pregnant women and immuno-compromised individuals (*Bonthius, 2012*; *Price et al., 1988*) and this too should be examined.

Globally infectious diseases remain a major cause of mortality and morbidity resulting in more than 15 million deaths per annum (*Morens et al., 2004*). However it is clear that many of the pathogenic agents responsible still remain undiagnosed, even when they are associated with potentially life threatening conditions. To improve protection we need a better understanding of both the ecology and epidemiology of diseases if we are to succeed in preventing them. Our findings illustrate how a 'One Health' approach in a regional hot-spot for emerging infectious disease can lead to the detection and characterisation of viruses likely associated with previously unrecognized human infections.

## Materials and methods

### Wild rodent sampling

No species included in this study are considered to be endangered or threatened and none are included on either the CITES list or the Red List (IUCN). Approval notices for trapping and investigation of rodents were provided by the Ministry of Health Council of Medical Sciences, National Ethics Committee for Health Research (NHCHR) in Laos (number 51/NECHR) and by the Ethical Committee of Mahidol University, Bangkok, Thailand (number 0517.1116/661) and permission to trap in Cambodia was granted by the Cambodian Ministry of Environment. As Cambodia has no ethics committee overseeing animal experimentation, animals were treated in accordance with the guidelines of the American Society of Mammalogists, and within the European Union legislation guidelines (Directive 86/609/EEC). Rodents were trapped in the Cambodian provinces of Mondulkiri (12°120′N; 106°890′E) and Veal Renh (10°710′N; 103°820′E), the Thai provinces of Buriram (14.89 N; 103.01 E), Loei (17.39 N; 101.77 E) and Nan (19.15 N; 100.83 E) and the Laotian provinces of Champasak (15.12 N; 105.80 E) and Luang Prabang (19.62 N; 102.05 E) as part of the CERoPath (Community

Ecology of Rodent-borne Pathogens) project (www.ceropath.org). Sites representing a range of habitats with differing degrees of human disturbance were trapped at each location. Two sampling sessions were conducted at each locality with 30 lines of ten traps set over four nights during each session, amounting to 1,200 trap nights per session. Locally made, wire live-traps (approx 40*12*12 cm) were used at each locality and were baited with cassava, banana or sticky rice. Captured rodents were collected each day, humanely euthanized and blood pellet, serum and organs were harvested, stored temporarily in liquid nitrogen and then transferred to long-term storage at -80°C. Selected samples were subjected to RT-PCR (see below) for evidence of arenavirus infection.

## Sample preparation and molecular detection of mammarenaviruses

Rodent lung samples were homogenised using the MagNA Lyser instrument and bead system (Roche, Basel, Switzerland). Tissue samples were placed in tubes containing ceramic-beads and 500 µl of pre-chilled sterile 1X PBS, subjected to oscillation at 3500 rpm and the lysate centrifuged at 6,200rpm for 5 min on a bench-top centrifuge. The RNA from homogenised rodent lung samples, human sera and respiratory specimens was extracted using the QIAmp Viral RNA mini kit (Qiagen, Hilden, Germany) as per the manufacturer's protocol.

Rodent samples were screened using a one-step RT-PCR targeting a conserved 395bp region of the L segment, using primers designed for Old World mammarenavirus detection and described previously by Vieth et al. (*Vieth et al., 2007*). To allow greater sensitivity a semi-nested RT-PCR ('screening' RT-PCR) was developed for the human samples, using the Vieth et al. method as the first step, followed by a nested step using primer LVL3359plus (*Vieth et al., 2007*) and a primer designed against Cardamones variant of Wēnzhōu virus, CAMN1 (*Figure 5* and *Supplementary file 1A*). Briefly, 1 µL of the first round reaction was added to a PCR mix comprising 13 µl Superscript II 2X reaction buffer (Life Technologies, Carlsbad, USA), 1 µl each of primers LVL3359plus and CAMN1 (each at 10 µM) (*Table 1*), 1 µL Platinum Taq polymerase (Life Technologies, Carlsbad, USA) and molecular grade water to a volume of 25 µl. A product size of 302 bp was generated for the semi-nested assay and addition of this step increased sensitivity by 1 Log-fold (data not shown). In addition a quantitative real-time RT-PCR (qRT-PCR) was developed and carried out using the SuperScript III Platinum One-Step qRT-PCR Kit (Life Technologies, Carlsbad, USA) and optimized according to the manufacturer's instructions. A volume of 5 µl of RNA was mixed with 12.5 µl of 2X Reaction Mix (0.4 mM of each dNTP, 3.2 mM MgSO4), 2 µM of the forward primer ARV2747-F, 1 µM of each of the reverse primers ARV-2937-R and ARV-3012R, 1 µM of ARV-probe, 0.5 µl of RNasin Plus RNase Inhibitor (40 U/µl; Promega, Madison, USA) and 1 µl of SuperScript III Platinum One-Step qRT-PCR enzyme to a final reaction volume of 25 µL. Thermocycling conditions were as follows: reverse transcription at 42°C for 30 min, denaturation at 95°C for 2 min and fluorescence detection for 45 cycles of 95°C for 30 sec, 50°C for 30 sec and 72°C for 1 min. Amplification and detection was performed on a LightCycler 480 II system (Roche, Basel, Switzerland) and amplification curves with CT values >35 were considered negative with the threshold line placed above the background signal, intersecting the initial log phase of the curve. In order to quantify mammarenavirus RNA, 395 bp PCR products were generated by conventional RT-PCR using the degenerate primers ARV2747-F and ARV3012-R (*Figure 5* and *Supplementary file 1A*). The cloning process was performed using the TA cloning kit (Invitrogen, Carlsbad, USA) according to the manufacturer recommendations. Plasmids containing the sequence of Cardamones variant of Wēnzhōu virus were diluted (1 to 1,000,000 copies/reaction) and included in each series of quantitative real-time RT-PCR.

## Whole Genome Sequencing, PASC and phylogenetic analysis

To obtain enough material for full genome sequencing, filtered supernatant from homogenized lung (50 mg in 500 µl PBS) from an infected, wild-caught Pacific rat (C0649) and from an infected, wild-caught Savile's bandicoot rat (R5074) were each used to inoculate one adult male Wistar laboratory rat by intra-peritoneal injection (500 µl). Both rats were sacrificed at 7 days post inoculation (dpi) and organs collected (see below for methodology). Total RNA from homogenised lungs were extracted using standard Trizol reagent as per the manufacturer's instructions (Life Technologies, Carlsbad, USA) and confirmed mammarenavirus-positive by RT-PCR (see above). Synthesis of cDNA, amplification, Illumina sequencing (on HiSeq 2000) and bioinformatics analysis were performed as described previously (*Cheval et al., 2011*). Using the program SOAPaligner (http://soap.genomics.org.cn), the

mouse (*Mus musculus*) genome 'NCBI37/mm9 assembly' reference genome, was used to filter out rodent genome-derived reads. Remaining reads were assembled using 3 different softwares: Velvet (www.ebi.ac.uk/~zerbino/velvewww.ebi.ac.uk/~zerbino/velvet), SOAPdenovo (http://soap.genomics. org.cn) and CLC Genomics Workbench (www.clcbio.com). Overlapping contigs assigned to an 'arenavirus' taxonomy were merged together to form longer sequences.

Primers were designed based on the sequences generated by Illumina sequencing and used to obtain the coding complete genome sequences from two further isolates (C0617 and R4937; *Supplementary file 1A*). To obtain S segment data a previously published protocol designed to generate full S segment amplicons and clones (*Bowen et al., 2000*) was used in conjunction with a series of RT-PCRs using the primers designed in this study (protocols available on request) and Sanger sequencing of the resultant amplicons. L segment sequence data was obtained using the primers designed in this study in a series of overlapping one step RT-PCRs (protocols available on request) and Sanger sequencing. Segments were reconstructed in the software CLC Main Workbench 5.5 (CLC bio A/S, Aarhus, Denmark). All genome sequences were submitted to GenBank on 20 February 2013.

Coding complete sequences for both segments of each virus were loaded into the PASC tool, accessible at the National Center for Biotechnology Information (NCBI) website (http://www.ncbi. nlm.nih.gov/sutils/pasc/viridty.cgi?textpage=overview), and analysed using the default parameters.

The sequences of New World and Old World mammarenavirus reference strains were retrieved from GenBank. Including the Cardamones variant of Wēnzhōu virus and Loei River virus sequences a total of 28 and 36 mammarenaviruses coding sequences for the L and S segments respectively, were aligned using Muscle (*Edgar, 2004*). The nt and aa identity matrices of the coding sequences were calculated by pairwise comparisons using p-distance in MEGA 5.2 (*Tamura et al., 2011*). Identity matrices were calculated for the Zinc protein (Z) and L coding sequences of L segment and the NP and GPC coding sequences of S segment separately. Jmodeltest (*Posada, 2008*) was used to select the optimal evolution model by evaluating the selected parameters using the Akaike Information Criterion (AIC). Phylogenetic analyses were performed using maximum likelihood (ML) method available in Seaview version 4.2.5 (*Galtier et al., 1996*; *Gouy et al., 2010*) using the recommended model GTR+G+I for the coding sequences of L, NP and GPC gene separately. The robustness of nodes was assessed with 1000 bootstrap replicates.

## Preliminary experiments in suspected rodent hosts

All work with infected rodents was carried out in a Bio-safety level 3 animal facilities and adhered to standard European guidelines for animal ethics (an animal ethics committee does not exist for Cambodia). Animals were kept in polyester filter bonneted cages, within a laminar flow Bio-isolator unit. Manipulations were carried out in a class 2 biosafety cabinet inside the BSL3 facilities. Animals were inoculated by intra-peritoneal injection, with 500 µl (adults) or 100 µl (juveniles) volumes and negative control animals were mock-inoculated with sterile PBS. At the end of each experiment animals were euthanized by exposure to chloroform followed by cervical dislocation. Personal protective equipment worn by the laboratory personnel comprised also a 3M Jupiter Tyvek laminate cape powered respirator (3M Jupiter, Diegem, Belgium), disposable gowns, shoe covers, and latex rubber gloves. Randomisation and blinding were not performed.

Experiments were set up to test for: 1) the potential for infection and its estimated duration; 2) the potential for horizontal transmission; 3) the potential for vertical transmission. Animals used to estimate duration of the infection were housed individually. For horizontal transmission experiments, animals were housed in pairs (1 inoculated and 1 un-inoculated individual) and dams were housed with their pups for vertical transmission experiments and neonatal inoculations. As numbers of animals were limited, experiments used to approximately estimate the duration of infection and potential for horizontal and vertical transmission were conducted at a number of time points with the number of animals available (see *Supplementary file 1C*). All animals were confirmed negative for mammarenavirus infection by RT-PCR and IFA (see below for details of method) before starting experiments. At the end of each experiment, blood was collected by cardiac puncture and serum separated by centrifugation. Brain, peritoneum, bladder, spleen, liver, kidney, lung and heart were collected immediately after death. Sera were tested by IFA and where the volume of blood was sufficient by ELISA (see below) and organs were subjected to RT-PCR (*Vieth et al., 2007*) and qRT-PCR

for selected individuals. Animals were deemed to have been infected if they demonstrated sero-conversion and/or were positive for mammarenavirus RNA.

For electron microscopy lung tissue from infected animals were collected, cut into small pieces, fixed in 2.5% glutaraldehyde 0.1 mol/L phosphate (Sorensen's buffer, pH 7.4) and post-fixed in 2% osmium tetroxide solution. After dehydration and embedding in Epon, semi-thin sections were cut and stained with toluidine blue. Ultra-thin sections were stained with uranyl acetate and lead citrate and then examined with an electron microscope (Jeol 100 CX II). For histopathological analysis organ samples were fixed in RCL2 (Alphelys, Plaisir, France) for 24 hr then dehydrated with pure alcohol. Samples were embedded in low melting wax (polyethylene-glycol disteate; Sigma-Aldrich, St Louis, USA), sectioned (coronally; 5-μm thickness) and stained with hematoxylin-eosin, periodic acid-Schiff and reticulin. For chromogenic immunohistochemistry human serum from a patient positive by ELISA (see below) was incubated overnight at 4°C with rodent tissue sections, in a buffer containing 0.5% bovine albumin. Samples were then exposed to a biotinylated donkey anti-human secondary antibody at 1:200 (Jackson ImmunoResearch Laboratories, Inc. West Grove, PA) followed by streptavidin-conjugated to horse radish peroxidase revealed by the AEC, a red chromogen at 1:400 (Vector Lab, Burligame, CA, USA). Sections were counterstained with haematoxylin. Negative controls comprised tissue sections incubated with normal polyvalent human immunoglobulins at the same concentration as the human sera used for mammarenavirus detection and incubated in the absence of immunoglobulins.

## Mammarenavirus isolation

Attempts to isolate the mammarenavirus from infected rodents' organs was conducted using VERO (ATCC CRL-1586), MDCK (ATCC CCL-34) and BHK-21 (ATCC CCL-10) cell lines previously described to isolate other mammarenaviruses. Briefly, each organ was homogenized using MagNA lyser instrument (Roche, Basel, Switzerland) as describe above and the homogenate was diluted at 1/100 with filtered culture medium (DMEM; Sigma Aldrich, Steinheim, Germany) supplemented with 5% of foetal calf serum (GIBCO) and 1% of a solution of penicillin-streptomycin (10000 units – 10 mg/mL; Sigma Aldrich, Steinheim, Germany) before inoculation onto each cell line. The cells were incubated at 37°C in a 5% $CO_2$ atmosphere for 7 days and the process was repeated 3 times. At each passage, the presence of the virus was detected by mammarenavirus qRT-PCR as described above. The presence of Mycoplasma contamination was not systematically checked before inoculation.

## Mammarenavirus IFA

Serological screening of experimentally infected rodents was initially performed by immune fluorescence assay (IFA) using LCMV-infected cells due to their availability and because this assay can detect numerous Old World mammarenavirus species. Sera were tested in pooled batches of five and sera from positive pools were then re-tested individually. Sera were diluted to 1:20 in PBS, added in 30 μl volumes to each well and placed at 37°C for 1 hr. Sera were aspirated from each well and slides were then washed three times in sterile PBS for 5 min each, rinsed in distilled water for 1 min, then allowed to air-dry. A secondary antibody (Sigma, anti-rat IgG FITC, conjugated in goat) at a dilution of 1:50 was then added in 30ul volumes to each well. Slides were placed at 37°C for 1 hr, then washed and rinsed as before. Once dry, PBS containing 30% glycerol was added in small quantities to each slide and a cover-slide placed firmly on top. Slides were visualized with a UV microscope.

## Wēnzhōu virus IgG ELISA

As the lack of a tissue culture isolate precluded the production of more IFA slides and in an attempt to produce a test more specific to Wēnzhōu virus, once the coding complete genome sequence of Cardamones variant of Wēnzhōu virus was obtained, a Sandwich ELISA (enzyme-linked immunosorbent assays) based on the nucleoprotein gene was developed. In order to do this, the nucleoprotein gene of this virus (GenBank accession number KC669696) was cloned into the plasmid pIVEX-MBP, and transformed in BL21 AI E. coli. After incubation and cell lysis, the lysate was purified by IMAC (Immobilized metal ion affinity chromatography; HisPur Cobalt Resin and purification kit, Thermo Scientific, Waltham, USA) to separate the fusion protein, (NP-ARV), from the host proteins. The fusion protein was cut at the TEV site, and a second IMAC purification was used to separate NP-ARV from

6His-MBP and 6His-TEV. SDS-PAGE demonstrated that NP-ARV was in the non retained fraction and that the purity was 83.3%. The protein was then concentrated on cellulose membrane (Amicon, Millipore) to a final concentration of 0.8 mg/ml. To obtain monoclonal antibodies (MAbs), BALB/c mice were immunized six times subcutaneously with 10 µg of purified NP-ARV antigen. The first immunization was carried out with complete Freund adjuvant (CFA) and the second two weeks later with incomplete Freund adjuvant (IFA). The MAbs exhibited different affinities for different epitopes: G14-95 with very low affinity (<10–7M) and H24-20 with good affinity (1.10–10M).

Sandwich ELISAs were conducted using MAb H24-20. This MAb was diluted in dilution buffer (1X PBS, 1% skimmed milk powder) and added to 96-well NUNC-immuno plates with a Polysorp surface (Nunc, Roskilde, Denmark) at 1 ug/ml per well. Plates were incubated at 4°C overnight, then washed four times with wash buffer (1X PBS, 0.05% Tween 20). Plates were blocked with blocking buffer (1X PBS, 5% skimmed milk powder) for 1 hr at 37°C then washed four times with wash buffer. A 100 µl volume of NP-ARV diluted to 0.8 ug/ml in PBS 1X was added per well and incubated for 1 hr at 37°C. Plates were then washed four times with wash buffer. Sera (human or rodent) diluted 1:100 in dilution buffer were added to wells in 100 µl volumes, incubated for 1 hr at 37°C, and then washed seven times with wash buffer. Peroxidase-labelled goat anti-human IgG (Biorad, Hercules, USA) or peroxidase-labelled goat anti-rat IgG (KPL catalog n# 04-16-02, Gaithersburg, MD, USA) was diluted 1:10,000 in dilution buffer and 100 µl added to each well. Plates were incubated for 1h at 37°C then washed four times with wash buffer. The substrate solution TMB (Tetramethyl Benzidine; KPL catalog # 50-76-01) was added in 100 µl volumes per well. The reaction was stopped by adding 100 µl of 1N $H_2SO_4$ and plates were measured at an absorbance of 450 nm. The means and standard deviations for each plate were calculated using 4 negative control serum samples. The cut-off value for the assay was defined as the mean plus three standard deviations (sd) following the formula: cut-off = mean + 3 sd + 10%. Each sample was tested twice in independent assays and when the results led to different conclusions, the sample was retested a third time. To verify the absence of cross-reactivity, human serum positive for anti-LCMV IgM and IgG (CDC, US) was tested in four independent series and found to test consistently negative for anti-Wēnzhōu IgG detection. Three positive controls (OD varying from low to high) were included in replicates in each plate and the intra-assay variation was considered as acceptable when the ODs of the positive controls were maintained within the ranges of their expected standard deviation values.

Seroconversion was defined as when the acute sample was negative and the convalescence sample of the same patient was positive, or when the convalescent sample demonstrated an increase of 50% or more of the OD value of the acute sera.

## Comparison of Mammarenavirus IFA and Wēnzhōu virus IgG ELISA

A total of 372 human serum samples and 7 experimentally infected rodent serum samples (*Supplementary file 1F and 1J*) were tested by both IFA and ELISA. The number of samples detected positive and negative by each test were used to calculate coefficients of correlation.

### Human samples

Human serum, respiratory and cerebro-spinal fluid samples used in this study were collected previously by the Institute Pasteur in Cambodia (IPC) within the framework of several research projects approved by the Cambodian National Ethics Committee for Health Research (NECHR) and stored in IPC's biobank. At the time of sample collection, a written consent form from each patient or their legal guardian was obtained and the NECHR also specifically authorized the use of these stored specimens for the purpose of the present study (NECHR No. 0205). The use of samples collected for dengue and influenza national surveillance and the use of stored and prospectively collected respiratory samples for the negative control groups were all approved by the NECHR. All the samples were anonymised for the purpose of this study. Samples were tested using the Wēnzhōu virus IgG ELISA or by semi-nested RT-PCR dependent on sample type (see above).

Eight groups of human samples were tested for evidence of mammarenavirus infection (*Table 3*). Group 1 (IgG_Dengue-Influenza-like_group 1_eLife_edit.xlsx in *Blasdell et al., 2016*) comprised acute and convalescent sera from 510 individuals with dengue- or influenza-like illness randomly selected between 2005 and 2010, originating from Kampong Cham and 11 other provinces. Paired sera comprised acute samples collected before day 4 of fever and convalescent samples collected 7

to 10 days after the onset of fever. Out of the 510 patients tested, only a single sample collected during the acute phase of the febrile episode was available for 98 patients, whilst only a single serum sample collected during the convalescent phase was available for 214 individuals. Paired serum samples, one collected during the acute phase and one during the convalescence phase were available from a further 198 patients. Group 2 (IgG_Healthy individual_group 2_eLife_edit. xlsx in *Blasdell et al., 2016*) consisted of sera collected from healthy individuals in a community based dengue seroprevalence study in 2009 in Kampong Cham province. The samples in both groups were tested by Wēnzhōu virus IgG ELISA. Groups 3 to 8 (PCR_Meningo-encephalitis patients_group 3_eLife_edit.xls, PCR_group 4-8_eLife_edit.xlsx in *Blasdell et al., 2016*) were all tested for mammarenavirus RNA by 'screening' semi-nested RT-PCR and qRT-PCR.

Group 3 comprised 200 patients hospitalized for meningo-encephalitis and who tested negative by PCR and/or serology for the main etiologies of central nervous systems infections usually observed in the country (Japanese encephalitis virus, dengue viruses, chikungunya virus, herpes simplex virus 1, influenza A virus, enteroviruses, Nipah virus, measles virus, mumps virus, rubella virus, *Streptococcus pneumoniae, Streptococcus suis, Haemophilus influenzae, Neisseria meningitidis, Orientia tsutsugamushi*). Group 4 consisted of randomly selected sera, obtained during the acute febrile phase from 253 patients. These patients, who originated from different parts of Cambodia, presented between 2009 and 2011 with signs and symptoms suggestive of dengue fever, dengue hemorrhagic fever and dengue shock syndrome. However all tested negative for dengue, Japanese encephalitis and chikungunya virus infections by RT-PCR and serology (in-house MAC-ELISA and hemagglutination-inhibition assay using antigens derived from the three arboviruses listed above, as described previously [*Andries et al., 2012*]). Group 5 comprised nasopharyngeal swab samples from 720 individuals presenting with an influenza-like syndrome. This group was further sub-divided into two sub-groups. Subgroup 5a included 328 individuals who tested negative for influenza A and B viruses, respiratory syncytial virus and human metapneumovirus by multiplex RT-PCR (*Buecher et al., 2010*; *Arnott et al., 2011*), whilst the 392 individuals in subgroup 5b tested positive for one of these viruses. Group 6 consisted of nasopharyngeal swabs from 279 individuals hospitalised with acute lower respiratory tract infections. Two negative control groups were also tested for mammarenavirus RNA. Group 7 comprised influenza A-negative nasopharyngeal specimens obtained from 266 apparently healthy individuals who had had contact with patients with H5N1 infections, while group 8 included nasopharyngeal swabs collected from 238 randomly selected apparently healthy volunteers seeking anti-rabies vaccination at IPC.

## Statistics used

All statistical analyses were performed using Stata/SE version 12.0 (StataCorp, TX, USA). Significance was assigned at $p < 0.05$ for all parameters and 95% of confidence interval was used. Categorical variables between groups were compared by Chi2 test or Fisher's exact test and t-test or Wilcoxon-Mann-Whitney test were used for continuous variables.

## Acknowledgements

IFA slides were kindly prepared and provided by Olli Vapalahti, Tytti Manni, Liina Voutilainen and Heikki Henttonen (Viral Zoonoses Research Group, Haartman Institute, Univ. Helsinki/Finnish Forest Research Institute, Finland). Arenavirus nucleoprotein (NP ARV) was kindly prepared by Pierre Beguin and Jacques Bellalou (Institut Pasteur, PF5 plate-form, Paris, France). The anti-NP monoclonal antibodies G14-95 and H24-40 were produced by Farida Nato (Institut Pasteur, PF5 plate-form, Paris, France). We are also very grateful to Amanda Bolanos-Gutierrez, Christopher Gorman, Sirenda Vong, Justine Cheval, Huy Rekol, Ngan Chantha, Arnaud Tarantola, Sok Touch, Ly Sovann for their kind assistance in this collaboration.

## Additional information

### Competing interests

PB: Currently an employee of GSK Vaccines R&D in Asia-Pacific region. The other authors declare that no competing interests exist.

## Funding

| Funder | Grant reference number | Author |
| --- | --- | --- |
| Agence Nationale de la Recherche | ANR07 BDIVLast 012 | Kim R Blasdell |

The funders had no role in study design, data collection and interpretation, or the decision to submit the work for publication.

## Author contributions

KRB, VDu, Conception and design, Acquisition of data, Analysis and interpretation of data, Drafting or revising the article; ME, FC, Acquisition of data, Analysis and interpretation of data, Drafting or revising the article, Contributed unpublished essential data or reagents; SL, Acquisition of data, Analysis and interpretation of data; VH, Conception and design, Acquisition of data, Analysis and interpretation of data, Contributed unpublished essential data or reagents; VDe, Analysis and interpretation of data, Drafting or revising the article; SM, Conception and design, Drafting or revising the article, Contributed unpublished essential data or reagents; PB, Conception and design, Acquisition of data, Analysis and interpretation of data, Drafting or revising the article, Contributed unpublished essential data or reagents

## Author ORCIDs

Serge Morand, http://orcid.org/0000-0003-3986-7659
Philippe Buchy, http://orcid.org/0000-0003-1372-3008

## Ethics

Human subjects: Human serum, respiratory and cerebro-spinal fluid samples used in this study were collected previously by the Institute Pasteur in Cambodia (IPC) within the framework of several research projects approved by the Cambodian National Ethics Committee for Health Research (NECHR; approval:NECHR No. 0205) and stored in IPC's biobank. At the time of sample collection, a written consent form from each patient or their legal guardian was obtained and the NECHR also specifically authorized the use of these stored specimens for the purpose of the present study. The use of samples collected for dengue and influenza national surveillance and the use of stored and prospectively collected respiratory samples for the negative control groups were all approved by the NECHR. All the samples were anonymised for the purpose of this study.

Animal experimentation: All work with infected rodents was carried out in a Bio-safety level 3 animal facilities and adhered to standard European guidelines for animal ethics (an animal ethics committee does not exist for Cambodia).

# Additional files

## Supplementary files

• Supplementary file 1. (A) Details of primers used for diagnostic RT-PCRs and genome sequencing. (B) Animals tested for arenavirus RNA by species and site, with number of positives shown in bold. (C) Details of rodent samples positive for arenavirus infection by screening RT-PCR . (D) Details of animals used for Cambodian virus infections. (E) Detailed IgG ELISA results for 7 patients with seroconversion. (F) Comparison between IFA and IgG ELISA results in human sera. (G) Arenavirus infections in patients with ILI symptoms who tested negative for 4 common respiratory viruses versus patients with ILI symptoms who tested positive and control group of healthy individuals. (H) Statistic analysis by age between respiratory illness group and healthy control group. (I) Multiple alignment of the sequences of the amplicons obtained by PCR for 3 patients. (J) Results of IgG ELISA in experimentally-infected rodents.

## Major datasets

The following dataset was generated:

| Author(s) | Year | Dataset title | Dataset URL | Database, license, and accessibility information |
|---|---|---|---|---|
| Kim R Blasdell, Veasna Duong, Marc Eloit, Fabrice Chretien, Sowath Ly, Vibol Hul, Vincent Deubel, Serge Morand, Philippe Buchy | 2016 | Data from: Evidence of human infection by new arenaviruses endemic to Southeast Asia | http://dx.doi.org/10.5061/dryad.9rh6v | Available at Dryad Digital Repository under a CC0 Public Domain Dedication |

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
