## [Decision Letter]

Thank you for submitting your work entitled "Evidence of human infection by new arenaviruses endemic to Southeast Asia" for consideration by *eLife*. Your article has been reviewed by Jens Kuhn and Peter Horby, and the evaluation has been overseen by Simon Hay as the Reviewing Editor and Richard Losick as the Senior Editor.

The reviewers have discussed the reviews with one another and the Reviewing Editor has drafted this decision to help you prepare a revised submission. You will see that we have garnered two very detailed and useful reviews that will help you improve the manuscript. Please run systematically through all comments and respond fully to the reviewers.

*Reviewer #1:*

"Evidence of human infection by new arenaviruses endemic to Southeast Asia" by Blasdell et al. provides the fundamental biological insight that at least two mammarenaviruses other than LCMV are endemic to Southeastern Asia and that one of them may infect people on a regular basis. One virus, Loei River virus is a new discovery; the second virus, which is associated with human disease, has recently been described in Eastern Asia, but until now was not considered a human pathogen. These findings are of considerable importance as they put mammarenaviruses on the list of possible etiological agents of orphan diseases.

In the first paragraph of the Introduction and in the last paragraph of the subsection “Identification of two arenaviruses in Southeast Asia”: the authors need to decide (using scientific methods) what they have discovered. If the virus they have discovered is a genetic variant of Wēnzhōu virus, then the virus needs to be called Wēnzhōu virus, not Cardamones virus (instead it would be the Cardamones variant or isolate of Wēnzhōu virus – note that publication takes precedent in nomenclature, not sequence deposition in GenBank). I see no problem with replacing "Cardamones" throughout with "Wēnzhōu".

If the new virus is clearly distinct from Wēnzhōu virus, then the name Cardamones virus could be justified (and a unique abbreviation should be introduced). If so, then the authors need to determine whether Cardamones virus belongs to the same species as Wēnzhōu virus (the recently proposed and by now ICTV-accepted *Wēnzhōu mammarenavirus*), i.e. whether this species will have two members (Cardamones virus and Wēnzhōu virus), or whether Cardamones virus needs to be classified in its own species.

To come to a decision, the authors should follow the updated guidelines published by the ICTV *Arenaviridae* Study Group (not the 9th ICTV Report as they have done): PMID: 25935216. Most important would be the placement of both novel viruses in PASC (S and L segment) – the results of which should be described in the manuscript. The last paragraph of the subsection “Identification of two arenaviruses in Southeast Asia” should then be updated accordingly and the reference be replaced (or PMID: 25935216 added in addition to the 9th ICTV Report).

In the last paragraph of the subsection “Human arenavirus infections in Cambodia”: what was the closest hit when the 302bp piece was BLASTed? That is, how specific is that piece truly for Wēnzhōu virus? The results of such a BLAST should be presented, as it is this small piece that is used to convince the reader that human infections are occurring.

The manuscript lacks precision and needs to be edited.

*Reviewer #2:*

General assessment:

This is an interesting manuscript that describes novel findings. It describes efforts to detect Old World arenaviruses in wild rodent species in Thailand, Cambodia, and Laos. Whilst I am not an expert in the laboratory aspects described, the successful experimental infection of animals using lung tissue that was positive for arenavirus RNA and the accompanying RT-PCR, serology and electron microscopy findings support the conclusion that the wild rodents were infected with an arenavirus. The virus is termed Cardamones virus in the manuscript but it is unclear that the virus represents a novel strain deserving of a new name. The failure to culture the virus is disappointing and the potential reasons for this require some discussion.

The manuscript then describes efforts to seek evidence of human infection with Cardamones virus. The evidence of human infections is based on serology and RT-PCR findings. The data presented do provide evidence of human infection with an arenavirus in Cambodia however the causal association between virus detection and illness is not wholly convincing.

Substantive concerns:

Whether the serological results indicate the infecting arenavirus is Cardamones virus infection, or another related and cross-reactive arenavirus, is uncertain, but this is addressed by the authors. Since serology results are a large component of the evidence for human infections with an arenavirus, more supportive data on the specificity of the serological assay would be helpful. Seroepidemiological surveys of novel viruses should ideally compare the seroprevalence in exposed populations to the seroprevalance in populations who have likely not been exposed to the animal reservoir and should report efforts to validate the assay in virologically-confirmed cases. The limitations of the validation of the specificity of the ELISA should be addressed in more detail and more clarity about the serology methods would be helpful:

a) What was the inter-assay correlation of the IFA and ELISA? b) What was the intra-assay reproducibility of the replicate independent IgG assays described in the subsection “Cardamones virus IgG ELISA”? Were acceptable replication parameters set?c) Please describe the definitions of 'seropositive' and 'seroconversion' more explicitly.

The serology positive rates and mean age of positive cases were not significantly different between healthy individuals and patients with a febrile illness, which leaves some doubt about the causal association between arenavirus antibody detection and illness. The seroconversions in unwell individuals is however an indicator of a possible causal link. Data on the interval between acute and convalescent samples for these seven cases should be presented. Presentation of the serology data could be improved – e.g. how were the +ve samples distributed by acute / convalescent samples?

Arenavirus PCR +ve respiratory samples were identified in 6 of 999 subjects with an acute respiratory illness compared to zero of 504 healthy controls. However, the control group was significantly older (28 vs. 16.7 years). This again leaves some doubt about the causal association between arenavirus detection and respiratory illness, since the statistical association may be confounded by age. This is particularly important since the highest antibody prevalence was in the 6-10 year old age group. I would suggest an age-stratified analysis of the PCR data if feasible.

---

## [Author Response]

Reviewer #1:

"Evidence of human infection by new arenaviruses endemic to Southeast Asia" by Blasdell et al. provides the fundamental biological insight that at least two mammarenaviruses other than LCMV are endemic to Southeastern Asia and that one of them may infect people on a regular basis. One virus, Loei River virus is a new discovery; the second virus, which is associated with human disease, has recently been described in Eastern Asia, but until now was not considered a human pathogen. These findings are of considerable importance as they put mammarenaviruses on the list of possible etiological agents of orphan diseases.

In the first paragraph of the Introduction and in the last paragraph of the subsection “Identification of two arenaviruses in Southeast Asia”: the authors need to decide (using scientific methods) what they have discovered. If the virus they have discovered is a genetic variant of Wēnzhōu virus, then the virus needs to be called Wēnzhōu virus, not Cardamones virus (instead it would be the Cardamones variant or isolate of Wēnzhōu virus – note that publication takes precedent in nomenclature, not sequence deposition in GenBank). I see no problem with replacing "Cardamones" throughout with "Wēnzhōu".

If the new virus is clearly distinct from Wēnzhōu virus, then the name Cardamones virus could be justified (and a unique abbreviation should be introduced). If so, then the authors need to determine whether Cardamones virus belongs to the same species as Wēnzhōu virus (the recently proposed and by now ICTV-accepted Wēnzhōu mammarenavirus), i.e. whether this species will have two members (Cardamones virus and Wēnzhōu virus), or whether Cardamones virus needs to be classified in its own species.

To come to a decision, the authors should follow the updated guidelines published by the ICTV Arenaviridae Study Group (not the 9th ICTV Report as they have done): PMID: 25935216. Most important would be the placement of both novel viruses in PASC (S and L segment) – the results of which should be described in the manuscript. The last paragraph of the subsection “Identification of two arenaviruses in Southeast Asia” should then be updated accordingly and the reference be replaced (or PMID: 25935216 added in addition to the 9th ICTV Report).

We are very grateful for all these comments and recommendations of the reviewer. We have now performed PASC analysis on both the Loei River virus sequences and the Cambodian sequences. Alongside the other species demarcation criteria stipulated by the ICTV, PASC analysis of Loei River virus indicates that this virus represents a novel mammarenavirus. PASC analysis of the Cambodian virus suggests that this virus belongs to the same species as Wēnzhōu virus, and this is supported by nucleoprotein amino acid sequence identities (<12%). However, as the identities between the Chinese and Cambodian sequences were relatively low by intra-species standards and because they were identified in geographically distinct regions, we propose that the Cambodian sequences represent a variant of Wēnzhōu virus, with the proposed name Cardamones. The name Cardamones virus has been replaced with Wēnzhōu virus throughout the text, or referred to as the Cardamones variant of Wēnzhōu virus. Although the Cardamones variant was identified in rodents of a species (*R. exulans*) not previously associated with Wēnzhōu virus, in addition to rodents of the species *R. norvegicus*, which has been, we don’t believe this supports the case for the Cardamones virus as a separate species. The previous study by Li et al., 2014 demonstrated that Wēnzhōu virus is clearly capable of infecting rodents of a range of species, including several *Rattus* species, so it’s presence in another one that is sympatric to *R. norvegicus* fits well with its seemingly promiscuous nature. The PASC results and discussion around the categorization of the Cambodian virus have been included in the Results and Discussion (see also Table 2 and [Supplementary-material SD1-data]) and the reference to the ICTV requirements updated as suggested.

In the last paragraph of the subsection “Human arenavirus infections in Cambodia”: what was the closest hit when the 302bp piece was BLASTed? That is, how specific is that piece truly for Wēnzhōu virus? The results of such a BLAST should be presented, as it is this small piece that is used to convince the reader that human infections are occurring.

The 340bp region from the Cambodian rodent samples (the 395bp region but excluding primer sequences) demonstrated 91% similarity to Wēnzhōu virus in BLAST analysis. The 302bp region from the human samples demonstrated 98-99% similarity to the sequences from the Cambodian rodent samples in pairwise alignment analysis, and 89-91% similarity to Wēnzhōu virus in BLAST analysis. This has now been included in the Results section.

*The manuscript lacks precision and needs to be edited.*

This has now hopefully been addressed.

Reviewer #2:

General assessment:

*This is an interesting manuscript that describes novel findings. It describes efforts to detect Old World arenaviruses in wild rodent species in Thailand, Cambodia, and Laos. Whilst I am not an expert in the laboratory aspects described, the successful experimental infection of animals using lung tissue that was positive for arenavirus RNA and the accompanying RT-PCR, serology and electron microscopy findings support the conclusion that the wild rodents were infected with an arenavirus. The virus is termed Cardamones virus in the manuscript but it is unclear that the virus represents a novel strain deserving of a new name. The failure to culture the virus is disappointing and the potential reasons for this requires some discussion.*

A full description of the culture method has been now included into the Materials and methods section. Some discussion has also been included around the possible reasons for failing to isolate the virus. Although the authors are unsure as to exactly why this was the case, as the Chinese strains of Wēnzhōu virus were isolated in a cell line not available in our laboratory, a suggestion to perform future isolation attempts in this cell line has been included in the Discussion.

The manuscript then describes efforts to seek evidence of human infection with Cardamones virus. The evidence of human infections is based on serology and RT-PCR findings. The data presented do provide evidence of human infection with an arenavirus in Cambodia however the causal association between virus detection and illness is not wholly convincing.

We agree and have tried to tone down this aspect. We have tried to make it clear that though an arenavirus appears to be causing human infections in Cambodia, we cannot confirm its identity or whether it is truly causing disease.

Substantive concerns:

Whether the serological results indicate the infecting arenavirus is Cardamones virus infection, or another related and cross-reactive arenavirus, is uncertain, but this is addressed by the authors. Since serology results are a large component of the evidence for human infections with an arenavirus, more supportive data on the specificity of the serological assay would be helpful. Seroepidemiological surveys of novel viruses should ideally compare the seroprevalence in exposed populations to the seroprevalance in populations who have likely not been exposed to the animal reservoir and should report efforts to validate the assay in virologically-confirmed cases.

We agree that such surveys should compare exposed and unexposed populations. Unfortunately, we did not have access to samples from an unexposed population and no longer have the resources to perform additional surveys at this stage. We did try to include samples from foreigners in our survey, but only few were available and it was unknown how long these individuals had been in the region for, so some may have been exposed. We have stated this in the Discussion and have also stated that such a comparison should be performed in future studies that should be easier to perform and to get funded after the publication of this manuscript. We have also stated that isolation of virus from suspect disease cases should be made in order to aid confirmation of causality.

The limitations of the validation of the specificity of the ELISA should be addressed in more detail and more clarity about the serology methods would be helpful:

a) What was the inter-assay correlation of the IFA and ELISA?

The inter-assay correlation of the IFA and ELISA were tested on a relatively small number of samples, due to the availability and limited volumes of many samples and the limited number of IFA slides kindly provided to us by the Hartman institute. Most of the IFA slides were used to test the samples from experimentally infected rodents, with only a small number used to test human samples. The ELISA was developed later and used primarily on the human samples. In the previous version of the manuscript it was incorrectly stated in the Methods that IFA *and* ELISA were used to test the rodent sera. Only limited volumes were obtained for the majority of samples. The ELISA results for the extremely limited number of experimentally infected rat samples that we were able to use were however consistent with the results found by IFA ([Supplementary-material SD2-data]).

We added some statistical comparisons between the 2 assays and also further clarified the differences and the performances of the serological tests.

“A limited number of human samples (Total = 372, [Supplementary-material SD2-data]) were tested by both mammarenavirus IFA and IgG ELISA, demonstrating in general consistent results (coefficient of correlation of 0.565 with a p<0.001).[…] As for most serological assays, the precise causative agent in these seropositive patients can therefore not be identified conclusively.”

In addition, in a new series of experimental infections in rats (C649_A and C649_B) conducted later when there was no IFA slide left, ELISA was able, as expected, to detect IgG antibodies in those 2 PCR-positive animals starting at day 7 and day 15 respectively ([Supplementary-material SD2-data]) which suggests that the sensitivity of the ELISA is satisfactory.

b) What was the intra-assay reproducibility of the replicate independent IgG assays described in the subsection “Cardamones virus IgG ELISA”? Were acceptable replication parameters set?

The intra-assay coefficients of variability ranged from 0 to 13.5%.

We systematically included 3 positive controls (OD varying from low to high) in replicates in each plate and the intra-assay variation was considered as acceptable when the ODs of the positive controls were maintained within the ranges of their expected standard deviation values.

This has now been added into the Materials and methods section.

c) Please describe the definitions of 'seropositive' and 'seroconversion' more explicitly.

The cut-off threshold was calculated as follows: (mean of negative +3SD) +10% and is mentioned in the Materials and methods section.

Out of the 510 patients tested, only a single sample collected during the acute phase of the febrile episode was available for 98 patients, whilst only a single serum sample collected during the convalescent phase was available for 214 individuals. Paired samples, one collected during the acute phase and one during the convalescence phase was available from a further 198 patients. In this context seropositive is defined as when acute and/or convalescent samples returned a value above the cut-off of the assay. Seroconversion was defined as when the acute sample was negative and convalescence sample of the same patient was positive or when an increase of more than 50% of OD value between acute and convalescent sera was measured. This has now been added to the Methods.

*The serology positive rates and mean age of positive cases were not significantly different between healthy individuals and patients with a febrile illness, which leaves some doubt about the causal association between arenavirus antibody detection and illness. The seroconversions in unwell individuals is however an indicator of a possible causal link. Data on the interval between acute and convalescent samples for these seven cases should be presented. Presentation of the serology data could be improved – e.g. how were the +ve samples distributed by acute / convalescent samples?*

We agree with the reviewer that a prospective study that includes IgG arenavirus serology in addition to the assays testing for the other main etiologies of febrile illnesses in the region would provide much better evidences. Unfortunately, as stated earlier, the data reported here used archived samples as the discovery of these new arenaviruses and their involvement in human infection was not expected when the CEROPATH study (that aimed at describing the diversity of several pathogens in rodent species) was designed. We hope that the publication of this paper will help ourselves and other groups to design, fund and conduct such clinical prospective studies in a near future.

In regards to seroconversion, the interval between the acute and convalescent samples of the 7 cases (seroconversion in 3 and increase of OD value of >50% in 4 samples) ranged between 2 and 15 days with a mean of 7.42 days. The intervals of the 3 seroconverted samples were 6, 9 and 15 days. [Supplementary-material SD2-data] with details on interval of between acute and convalescent samples of the 7 cases was added.

Additional information on positive samples by acute or convalescent samples has been added to Table 2.

Arenavirus PCR +ve respiratory samples were identified in 6 of 999 subjects with an acute respiratory illness compared to zero of 504 healthy controls. However, the control group was significantly older (28 vs. 16.7 years). This again leaves some doubt about the causal association between arenavirus detection and respiratory illness, since the statistical association may be confounded by age. This is particularly important since the highest antibody prevalence was in the 6-10 year old age group. I would suggest an age-stratified analysis of the PCR data if feasible.

We thank the reviewer for this suggestion and have added [Supplementary-material SD2-data].

We agree that it is unfortunate that the healthy control group did not perfectly match the age distribution of the patient’s group but again this is due to the retrospective approach based on availabilities of clinical specimens that made it extremely difficult to achieve a perfect age match. If unfortunately the sample size of the young age control groups is too small to reach statistical significance, a significant association was still observed in the group of adults over 30 years as well as for the all-age population.